# Microglia are not protective against cryptococcal meningitis

Sally H. Mohamed[1], Man Shun Fu[1], Sofia Hain [1], Alanoud Alselami[1], Eliane Vanhoffelen[2], Yanjian Li[3], Ebrima Bojang[1], Robert Lukande[4], Elizabeth R. Ballou [5], Robin C. May [6], Chen Ding [3], Greetje Vande Velde [2] & Rebecca A. Drummond [1,6]✉

Microglia provide protection against a range of brain infections including bacteria, viruses and parasites, but how these glial cells respond to fungal brain infections is poorly understood. We investigated the role of microglia in the context of cryptococcal meningitis, the most common cause of fungal meningitis in humans. Using a series of transgenic- and chemical-based microglia depletion methods we found that, contrary to their protective role during other infections, loss of microglia did not affect control of *Cryptococcus neoformans* brain infection which was replicated with several fungal strains. At early time points post-infection, we found that microglia depletion lowered fungal brain burdens, which was related to intracellular residence of *C. neoformans* within microglia. Further examination of extracellular and intracellular fungal populations revealed that *C. neoformans* residing in microglia were protected from copper starvation, whereas extracellular yeast upregulated copper transporter CTR4. However, the degree of copper starvation did not equate to fungal survival or abundance of metals within different intracellular niches. Taken together, these data show how tissue-resident myeloid cells may influence fungal phenotype in the brain but do not provide protection against this infection, and instead may act as an early infection reservoir.

Tissue-resident macrophages populate every organ in the body and help maintain homoeostasis while also acting in an immune surveillance capacity to provide rapid responses to infection or injury. In the central nervous system (CNS), there are several populations of tissue-resident macrophages that associate with distinct anatomical locations, including the meninges (meningeal macrophages), blood vessels (perivascular macrophages) and the brain parenchyma (microglia)[1].

Microglia are the most numerous brain-resident macrophages. These cells derive from embryonic precursors and maintain their numbers by self-renewal, relying on signals via the CSF1R for survival[1].

Microglia are critical for proper brain development by helping to prune neuronal synapses and clear cellular debris. In adulthood, one of the major functions of microglia is immune surveillance and front-line defence against brain infection. The CNS is susceptible to infection with many pathogens including bacteria, viruses, parasites and fungi. Microglia are ideally situated to respond rapidly to pathogens as they invade the brain. For example, microglia are required for protective immunity against neurotropic coronaviruses by shaping CD4 T-cell responses within the brain, since their depletion results in uncontrolled viral load and accelerated mortality[2]. During infection with the parasite *Toxoplasma gondii*, microglia respond rapidly to produce the

[1]Institute of Immunology & Immunotherapy, University of Birmingham, Birmingham, UK. [2]Department of Imaging and Pathology, Biomedical MRI/MoSAIC, KU Leuven, Leuven, Belgium. [3]College of Life and Health Sciences, Northeastern University, Shenyang 110015 Liaoning, China. [4]Department of Pathology, College of Health Sciences, Makerere University, Kampala, Uganda. [5]MRC Centre for Medical Mycology, University of Exeter, Exeter, UK. [6]Institute of Microbiology & Infection and School of Biosciences, University of Birmingham, Birmingham, UK. ✉e-mail: r.drummond@bham.ac.uk

alarmin IL-1α, required to initiate immune cell infiltration and control of parasite infection[3].

CNS infections with fungi are much rarer than those caused by other microbes, but are difficult to treat and frequently lethal. During brain invasion by *Candida* fungi, microglia depend on CARD9 to activate antifungal immune responses such as the production of neutrophil-attracting chemokines[4]. Consequently, human CARD9 deficiency is a primary immunodeficiency disorder that leads to life-threatening CNS *Candida* infections[5].

While many pathogenic fungi are able to infect the human CNS, the most common species is *Cryptococcus neoformans*, leading to a disease called cryptococcal meningitis. Cryptococcal meningitis is a leading cause of death in HIV/AIDS patients and is an increasingly observed clinical complication in patients with underlying immune defects caused by iatrogenic interventions[6]. How microglia respond to this fungal infection is unclear. Some studies have observed uptake and survival of the fungus within microglia[7], while others have shown robust microglia activation leading to expression of MHC Class II and iNOS following infiltration by IFNγ-producing CD4 T-cells to the CNS[8] or following administration of immunotherapy[9]. Moreover, the microenvironment of the CNS and brain tissue has significant influence over *C. neoformans* morphology and growth responses. For example, *C. neoformans* is severely restricted of the micronutrient copper in the brain, but must deal with toxic copper levels in other organs such as the lung[10]. Whether tissue-resident macrophages drive these fungal adaptions by regulating access to micronutrients, and the impact of nutritional immunity to organ-specific immune responses, is not known.

We set out to understand the in vivo role of microglia during early infection with *C. neoformans* using a series of microglia depletion strategies. Our findings show that, contrary to other types of infections studied thus far, microglia do not provide protection against this fungal infection, but played significant roles in influencing the development of heterogeneity within fungal populations in this tissue.

## Results

### Microglia harbour intracellular fungi

*C. neoformans* can survive and replicate within macrophage phagosomes, aiding its evasion of the immune system and dissemination to the brain within monocytes[11,12]. We first characterised the association between *C. neoformans* and brain-resident and recruited inflammatory myeloid cells, using a green fluorescent reporter *C. neoformans* strain, which allowed us to track fungal uptake by different myeloid cell populations in the brain in vivo using flow cytometry (Fig S1). These experiments revealed that ~5% of microglia became infected with *C. neoformans* (Fig. 1a). However, microglia were the most numerous infected cell type within the brain, with significantly greater numbers of infected microglia detected compared to neutrophils, monocytes and inflammatory macrophages (Fig. 1b). When examined as a proportion of total infected myeloid cells in the brain, microglia accounted for nearly half of all fungal-infected cells (Fig. 1c). Analysis of fungal localisation by histology showed that yeast were growing in the brain parenchyma, in close association or intracellularly in host cells, and within areas of tissue damage with little association with host cells (extracellular growth) (Fig. 1d). Confocal microscopy revealed that microglia with internalised yeast had a rounded, amoeboid morphology typical of activation[1]. These infected cells were often found next to sites of tissue damage that housed large numbers of extracellular yeast (Fig. 1e). As infection progresses, large areas of tissue damage with extracellular yeast growth become more prominent (Fig. 1d, e). Indeed, quantification of the proportion of extracellular yeast (as determined by histology) was increased at day 6 post-infection compared to day 3 post-infection (Fig. 1f). We observed similar patterns of growth in the human brain. Pathology analysis of brain tissue from a patient who

died from HIV-associated cryptococcal meningitis had areas of extensive tissue damage with large numbers of extracellular yeast, and in other areas, yeast were observed residing within immune cells (Fig. 1g). Taken together, this data shows that an intimate relationship exists between tissue-resident microglia and *C. neoformans*.

### Brain macrophages are not required for controlling *C. neoformans* infection

While the role of monocytes and inflammatory macrophages in the control of *C. neoformans* infections has been intensely studied, the role of tissue-resident myeloid cells is less well understood. Since we had observed a close relationship between brain-resident myeloid cells and *C. neoformans* during early infection with increasing extracellular growth late post-infection, we decided to definitively determine the role of CNS-resident myeloid cells at various time points post-infection using in vivo mouse models. For that, we used several depletion strategies and assessed their effects on control of fungal brain infection. First, we generated *Cx3cr1*-Cre[ER]-iDTR[flox] animals in which all long-lived CX3CR1[+] cells in the brain are depleted following treatment with diphtheria toxin (Fig. 2a). Microglia highly express CX3CR1, and unlike other CX3CR1+ myeloid cells, microglia do not replenish from the bone marrow. Therefore, we can target expression of the diphtheria toxin receptor (DTR) to only long-lived cells by including a rest period in which peripheral immune cells turn over from the bone marrow (Fig. 2a). To examine the role of microglia during experimental *C. neoformans* infection, we used an intravenous infection model where yeast cells directly invade the CNS and bypass pulmonary immunity, allowing us to isolate host-fungal interactions within the CNS. Using *Cx3cr1*-Cre[ER]-iDTR[flox] mice, we found that depletion of CX3CR1[hi]CD45[int] microglia was maintained during fungal infection (Fig. 2b). Following *C. neoformans* challenge, we found that the depleted mice had significantly reduced brain fungal burdens but no difference in lung fungal burden (Fig. 2c), indicating that microglia depletion may have a protective effect. To confirm these findings, we employed an independent microglia depletion strategy using the CSF1R inhibitor, PLX5622 (Fig. 2e, f), which interrupts the CSF1R-dependent signalling required for microglia survival (Fig S2). We found that PLX5622-treated mice also had reduced fungal brain burdens compared to untreated controls early post-infection, whereas lung burdens remained unchanged (Fig. 2g). However, in contrast to the Cx3cr1-Cre[ER]-iDTR[flox] mice, we found no difference in fungal brain burdens at day 6 post-infection in PLX5622-treated mice (Fig. 2g). In both models, we found that microglia depletion had little or no impact on recruitment of inflammatory myeloid cells such as Ly6C[hi] monocytes and neutrophils to the brain (Fig. 2d, h). Taken together, these data show that microglia do not provide protection against *C. neoformans* infection, and that depletion of these cells may support fungal growth at early time points.

### Early fungal brain infection is primarily supported by microglia

PLX5622 treatment can also deplete other macrophages found within the CNS, including recruited inflammatory macrophages and CNS-resident meningeal macrophages[2,13] (Figs. 2h and S2). Moreover, the *Cx3cr1*-Cre[ER] line has been shown to additionally target non-microglia populations in certain contexts[14,15]. We hypothesised that differences in how these two models affected non-microglia populations may explain the different phenotypes we observed in control of *C. neoformans* infection at later time points post-infection. Therefore, we sought to generate a mouse model to enable specific depletion of microglia while leaving other CNS-resident and recruited inflammatory macrophages intact, to definitively determine the impact of microglia loss on control of *C. neoformans* brain infection.

*Sall1* is a signature microglia gene that is specifically expressed by microglia but not by other CNS-resident or

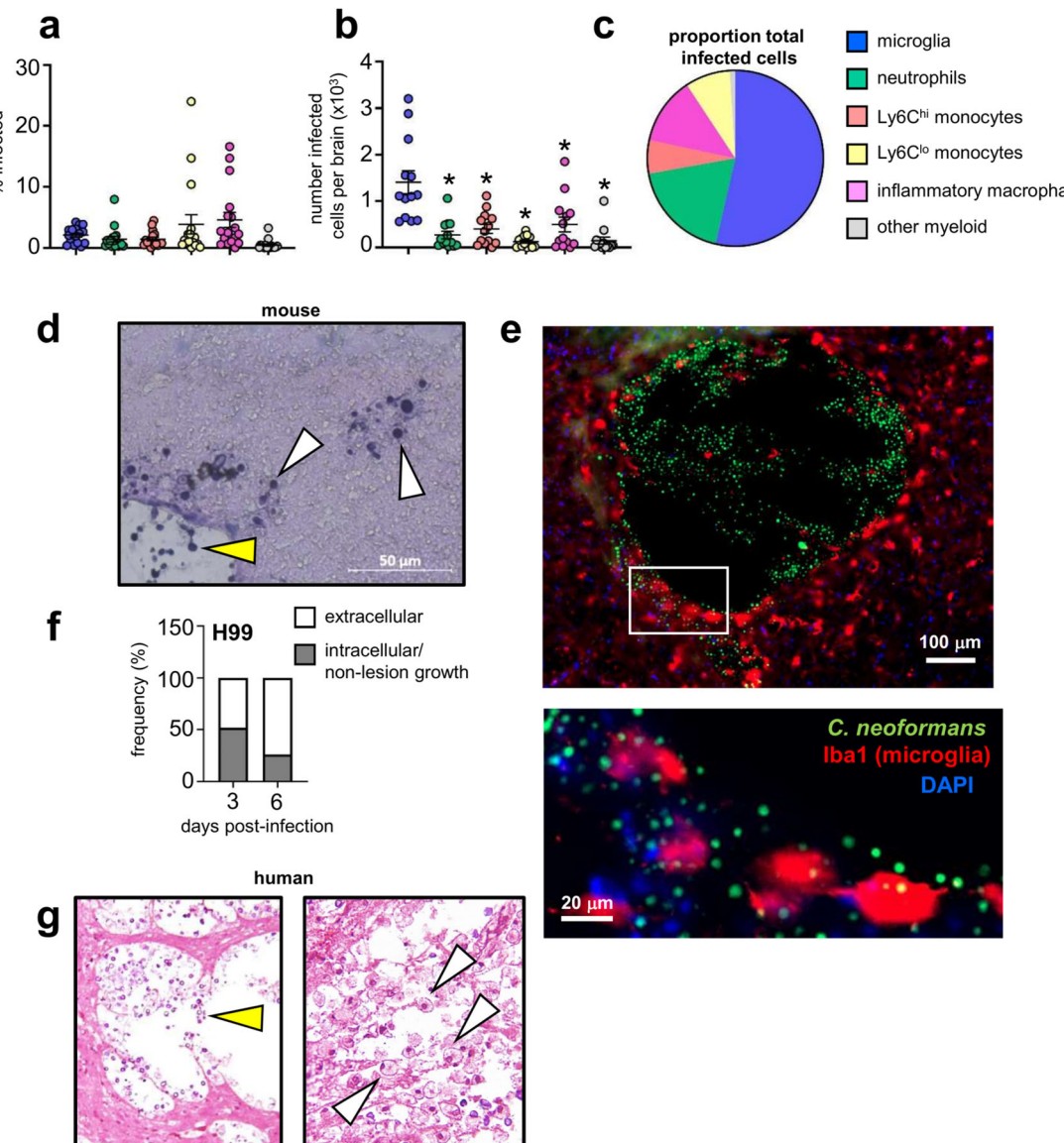

**Fig. 1 | Microglia are hosts to intracellular fungi. a** Frequency of *C. neoformans* infection within indicated cell types in the brain at day 7 post-infection. Data are pooled from 4 independent experiments (*n* = 17 mice). **b** Total number of infected cell types in the brain at day 7 post-infection. Data presented as mean +/- SEM and are pooled from 4 independent experiments (*n* = 13 mice) and analysed by one-way ANOVA with Bonferroni correction. *Adjusted *P*-value of <0.0001. **c** Proportion of indicated cell types within total infected cells at day 7 post-infection. An average value of 4 independent experiments for each cell type is shown. See Fig S1 for gating strategies. **d** Representative histology of mouse brain at day 6 post-infection, stained with Periodic-Acid Schiff (PAS). White arrows denote intracellular growing yeast, yellow arrows denote extracellular growing yeast. Similar results were observed across 3 brains analysed. **e** Confocal microscopy images of example tissue lesion with yeast and intracellularly-infected microglia. Mice were infected with GFP-expressing *C. neoformans* as before, and brains isolated at day 7 post-infection for analysis. Brain sections were stained with DAPI and anti-Iba1 to label microglia. GFP signal was used to identify yeast cells. Similar results were observed across 3 brains analysed. **f** Ratio of intracellular (grey bar) and extracellular (white bar) yeast counted across 3-5 sections from two mouse brains (at each time point), expressed as proportion of total counted yeast. **g** Histology of human brain, isolated at autopsy from a patient with HIV-associated cryptococcal meningitis, stained with PAS. White arrows denote intracellular growing yeast, yellow areas denote extracellular growing yeast. Similar results were observed in six sections analysed. Source data are provided as a source data file.

inflammatory macrophages[16]. We confirmed this specificity by generating *Sall1^CreER^R26^Ai14^* transgenic mice, in which only microglia became labelled with dTomato following tamoxifen treatment (Fig. 3a, b). Sall1-dependent labelling of microglia was stable during *C. neoformans* infection and remained specific to this cell type (Fig. 3c). Therefore, Sall1 is a reliable marker for tissue-resident microglia that can be used to specifically target these cells during experimental *C. neoformans* infection. We next crossed *Sall1^CreER^* animals with *Csf1r^flox^* transgenic mice to delete CSF1R specifically in microglia. We found that *Sall1^CreER^Csf1r^flox^*

mice had specific depletion of microglia but normal numbers of meningeal macrophages and other macrophage populations in the brain after tamoxifen treatment (Figs. 3d, e and S2). Importantly, these microglia-specific depleted mice had significantly reduced brain fungal burden at day 3 post-infection (Fig. 3f), in line with our other depletion models. At day 6 post-infection, there was no significant difference in fungal brain burden in microglia-depleted mice compared to wild-type littermate controls (Fig S3). Taken together, these data confirm that microglia do not provide protection against *C. neoformans* infection.

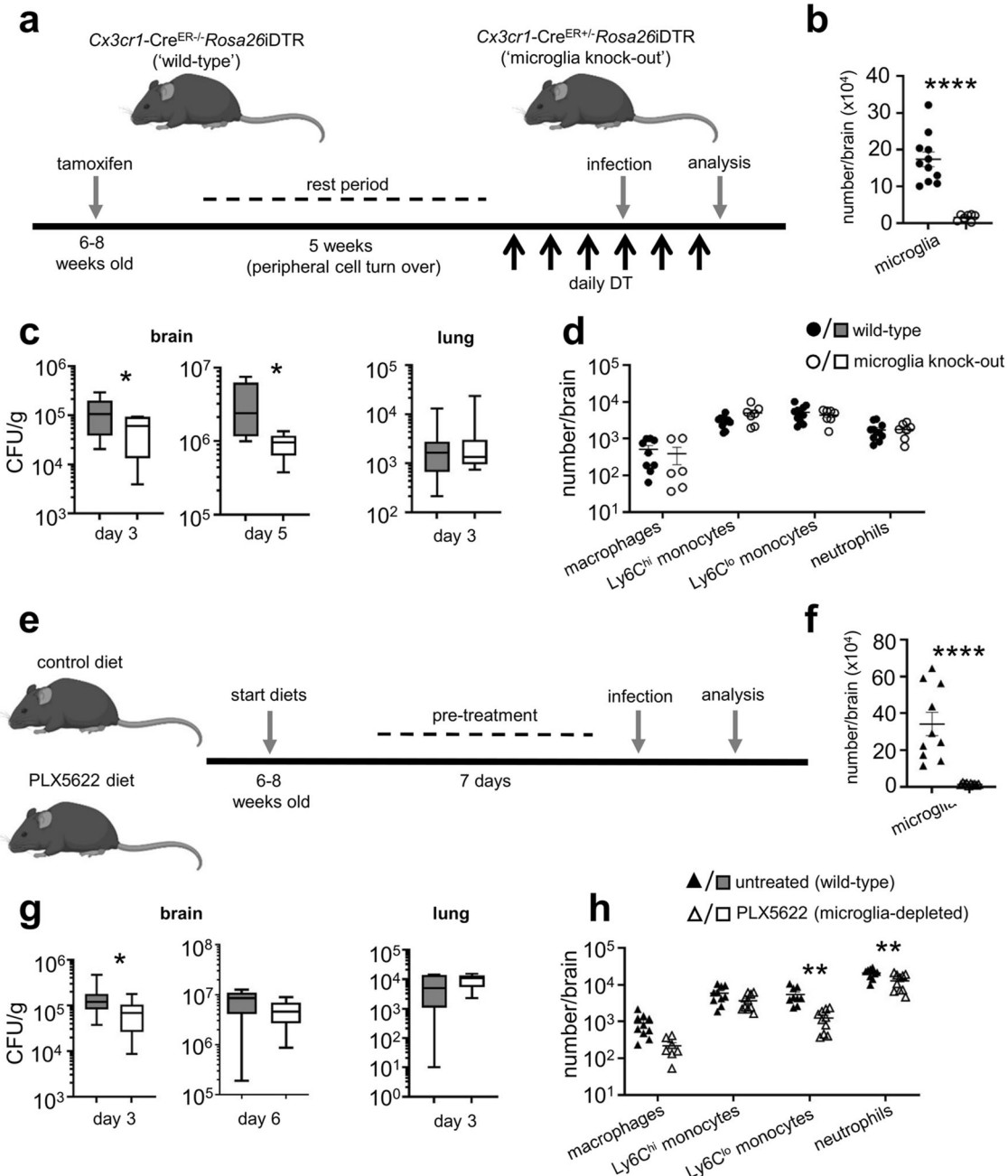

**Fig. 2 | Depletion of CNS-resident macrophages reduces fungal brain burden.**
**a** Schematic of diphtheria toxin-based depletion of brain-resident macrophages. *Cx3cr1*-Cre^ER mice were crossed with iDTR mice. Resulting Cre+ ('wild-type') and Cre- ('microglia knock-out') littermates are treated with tamoxifen to induce Cre expression, and left to rest for 5 weeks to enable turn-over of monocyte-derived macrophages prior to daily treatment with diphtheria toxin to initiate cell depletion before and during intravenous *C. neoformans* H99 infection. **b** Total number of microglia in wild-type (*n* = 11; black circles) and microglia knock-out (*n* = 7; open circles) brains at day 3 post-infection. Data are pooled from 2 independent experiments and analysed by unpaired two-tailed t-test. ****$P$ < 0.0001. **c** Fungal burdens in the brain ($P$ = 0.0441) and lung ($P$ = 0.7914) of wild-type (*n* = 11) and microglia knock-out (*n* = 7) mice at day 3 or 5 (*n* = 11 wild-type mice, *n* = 7 microglia knock-out mice; $P$ = 0.0109) post-infection. Data are pooled from 2 independent experiments and analysed by two-tailed Mann Whitney U-test. Box plots show median with 25%/75% percentiles and maximum and minimum values. *$P$ < 0.05. **d** Total number of indicated inflammatory cells in the brains of wild-type (*n* = 11) and microglia knock-out (*n* = 6) mice at day 3 post-infection. Data are pooled from 2

independent experiments. **e** Schematic of PLX5622 treatment. Wild-type C57BL/6 mice were fed either control diet or PLX5622 diet for 7 days prior to intravenous infection with *C. neoformans* H99. Diets were continued throughout infection. **f** Total number of microglia in untreated (*n* = 10; black triangle) and PLX5622-treated (*n* = 10; open triangle) brains at day 3 post-infection. Data are pooled from 2 independent experiments and analysed by unpaired two-tailed t-test. ****$P$ < 0.0001. **g** Fungal burdens in the brain ($P$ = 0.0186) and lung ($P$ = 0.4876) of untreated (*n* = 14 brain, *n* = 5 lung) and PLX5622-treated (*n* = 14 brain, *n* = 5 lung) mice at day 3 or day 6 (*n* = 9 per group; $P$ = 0.1081) post-infection. Data are pooled from 1 (lung) or 3 (brain) independent experiments and analysed by two-tailed Mann Whitney U-test. *$P$ < 0.05. Box plots show median with 25%/75% percentiles and maximum and minimum values. **h** Total number of indicated inflammatory cells in the brains of untreated (*n* = 10) and PLX5622-treated (*n* = 10) mice at day 3 post-infection. Data are pooled from 2 independent experiments and analysed by two-way ANOVA with Bonferroni correction ($P$ = > 0.99 macrophages, $P$ = 0.6948 Ly6C^hi monocytes, $P$ = 0.0456, Ly6C^lo monocytes, $P$ = 0.0001 neutrophils). Mouse icons created with Biorender.com. Source data are provided as a source data file.

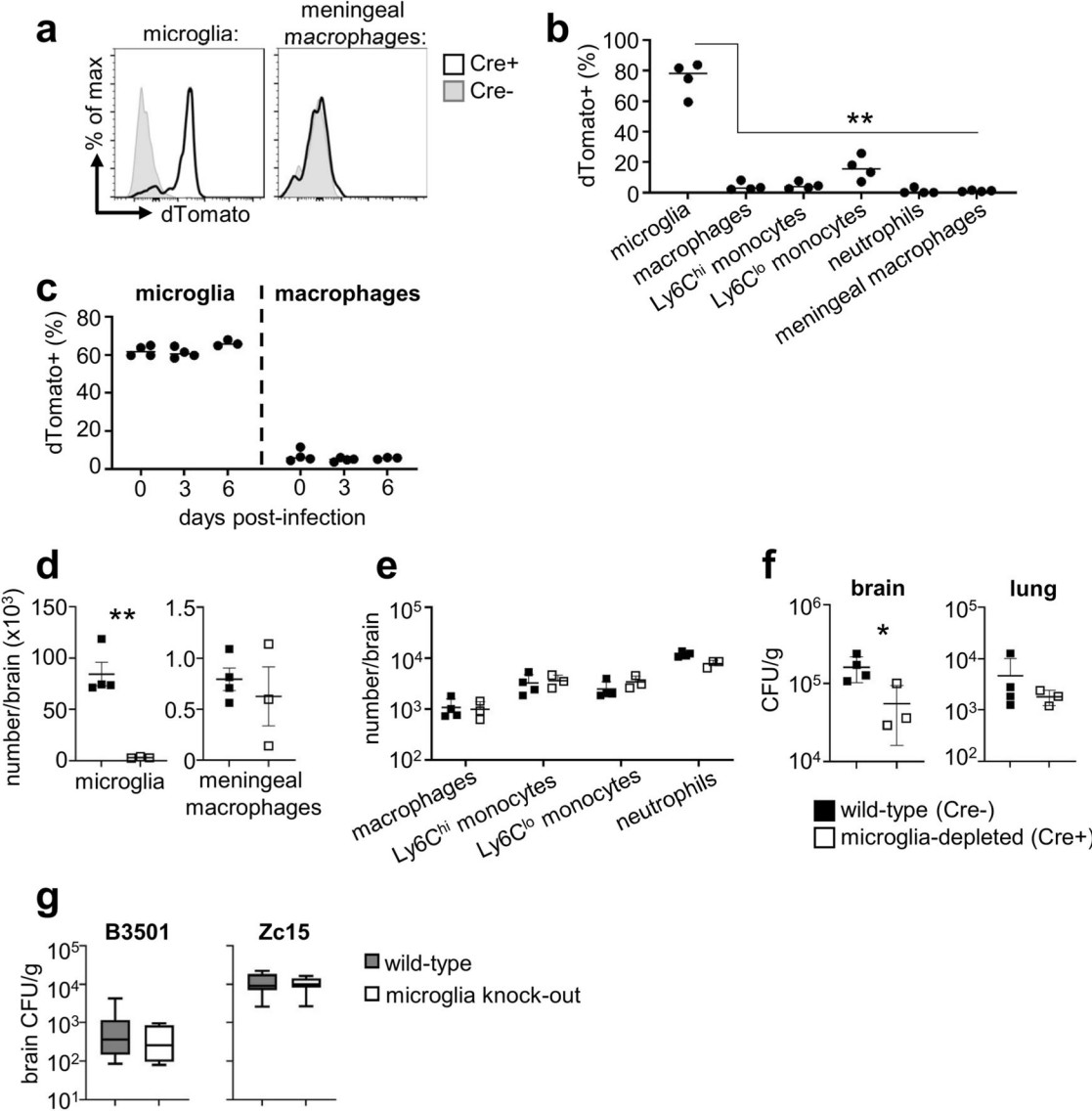

**Fig. 3 | Specific microglia depletion reduces brain fungal burden. a** *Sall1*-Cre[ER] mice were bred with *Rosa26*[Ai14] animals and treated with tamoxifen to activate Cre and dTomato expression. Microglia and meningeal macrophage expression of dTomato is shown for representative Cre+ and Cre- littermates. **b** Frequency of dTomato expression in the indicated cell populations in the brain in 4 Cre+ litter-mates. Data are from 1 experiment and analysed by one-way ANOVA comparing to microglia ($P = 0.0098$ macrophages, $P = 0.0035$ Ly6C[hi], $P = 0.0021$ Ly6C[lo], $P = 0.0047$ neutrophils, $P = 0.005$ meningeal). **$P < 0.01$. **c** Frequency of dTomato expression in microglia and brain macrophages in uninfected ($n = 4$) and infected ($n = 4$ mice at days 0 and 3, $n = 3$ mice at day 6) Cre+ mice. **d** Total number of microglia ($P = 0.0018$) and meningeal macrophages ($P = 0.5682$) in wild-type ($n = 4$; black square) and microglia-depleted ($n = 3$; open square) brains at day 3 post-infection. Data are representative of 2 independent experiments and analysed by

unpaired two-tailed t-test. **$P < 0.01$. **e** Total number of indicated inflammatory cells in the brains of wild-type ($n = 4$) and microglia-depleted ($n = 3$) brains at day 3 post-infection. Data are representative of 2 independent experiments. **f** Fungal burdens in the brain ($P = 0.0442$) and lung ($P = 0.7346$) of wild-type ($n = 4$) and microglia-depleted ($n = 3$) brains at day 3 post-infection. Data are representative of 2 independent experiments and analysed by unpaired two-tailed t-test. Data shown as mean +/− SEM. *$P < 0.05$. **g** Fungal brain burdens in wild-type (*Sall1*[CreER]*Csf1r*[flox]; Cre-negative littermates) and microglia-deficient (*Sall1*[CreER]*Csf1r*[flox]; Cre-positive lit-termates) mice at day 3 post-infection with *C. deneoformans* B3501 ($n = 10$ wild-type, $n = 7$ microglia-deficient) and *C. neoformans* Zc15 ($n = 6$ wild-type, $n = 8$ microglia-deficient). Data pooled from two independent experiments. Box plots show median with 25%/75% percentiles and maximum and minimum values. Source data are provided as a source data file.

## Microglia do not provide protection against a range of *C. neoformans* clinical isolates

Our experiments utilised the laboratory reference strain *C. neoformans* H99, a virulent serotype A strain that was originally isolated from a lymphoma patient at Duke University[17]. *C. neoformans* H99 has been widely used for infection studies and determining the mechanisms of intracellular invasion and residence, as well as generating fluorescent reporter strains and mutant libraries[17]. However, there is wide variation between clinical and environmental isolates of *C. neoformans*, as well as between laboratory reference strains[17]. These variations can have

significant impact on host immune responses and fungal morphology[18]. We therefore examined the role of microglia for con-trolling infection in mice infected with different isolates of *C. neofor-mans*. We first chose to examine *C. neoformans* B3501, which is commonly used for infection studies as a serotype D reference strain[19,20]. We found that microglia-deficient mice infected with B3501 had similar brain fungal burdens compared to microglia-sufficient controls (Fig. 3g). Since B3501 is a different serotype to H99, we next decided to test a recent clinical isolate of *C. neoformans*, Zc15, that is also serotype A but genetically distinct from H99[21]. Mice infected with

Zc15 had ~10 fold less fungal brain burden compared with H99 (compare Fig. 3g with Fig. 2c, g), indicating that Zc15 has reduced virulence in mouse models of *C. neoformans* infection. However, the absence of microglia did not affect control of infection in mice infected with Zc15 (Fig. 3g). Taken together, these results confirm that microglia are not protective for controlling *C. neoformans* infection across multiple strain backgrounds.

### Reduction of brain fungal infection in microglia-depleted mice is dependent on intracellular fungal growth

*C. neoformans* strain H99 has a profound ability to grow intracellularly within myeloid cells. Infection of microglia-deficient mice with this strain showed a reduction in fungal brain burden in multiple depletion models, but this was isolated to an early time point post-infection (Figs. 2 and 3). We hypothesised that this phenotype was the result of early intracellular growth of *C. neoformans* within microglia, particularly since we had observed an increased rate of extracellular fungal growth at later time points post-infection (Fig. 1f) and had found no significant difference in brain burdens of microglia-depleted mice at this time point in two of the models tested (Fig. 2g and S3). To determine the functional relevance of fungal intracellular survival within microglia at early time points post-infection, we used the *rdi1Δ* *C. neoformans* mutant (on the H99 background). This strain is unable to survive within macrophages but has normal growth rates in rich media[22]. Histological examination of mouse brains infected with the *rdi1Δ* mutant revealed primarily extracellular yeast in areas of tissue damage at 72 h post-infection, whereas wild-type yeast cells were both intracellular and extracellular at this time (Fig. 4a). Quantification of these images revealed that 51.9% of counted wild-type yeast were intracellular, compared to 8.6% for the *rdi1Δ* mutant (Fig. 4b). We infected mice that were either microglia-sufficient (untreated) or microglia-depleted (PLX5622) with wild-type or *rdi1Δ* *C. neoformans*, and then quantified brain fungal burden. Compared to wild-type *C. neoformans*, brain fungal burdens were significantly reduced with the *rdi1Δ* strain (Fig. 4c). Importantly, while microglia depletion caused a significant reduction in brain infection with the wild-type *C. neoformans*, there was no significant difference in brain fungal burdens between microglia-depleted mice and their untreated controls when infected with the *rdi1Δ* mutant (Fig. 4c). These data indicate that the reduction in brain fungal burden observed with microglia depletion is dependent on the ability of the fungus to survive intracellularly.

### *C. neoformans* is protected from copper starvation within microglia

Microglia provide protection against a variety of brain-invading pathogens, including other fungi[4]. The lack of an effect of microglia depletion on control of *C. neoformans* infection was therefore surprising, given that these cells become intracellularly infected with the fungus and are thought to be among the first responders to this infection within the brain. To better understand the relationship between *C. neoformans* and microglia, we examined the phenotype of fungal cells residing in microglia and compared to extracellular yeast populations. We decided to focus on nutrient acquisition pathways, since we hypothesised that microglia may be unable to restrict fungal access to micronutrients, thus providing a 'safe haven' for *C. neoformans* from starvation conditions found elsewhere in the brain which may explain why depletion of microglia at early time points reduced fungal brain infection.

Several micronutrients are restricted within the CNS including copper and iron, but whether CNS-resident macrophages participate in this restriction is not well understood. Indeed, copper is typically found bound within microglia, with extracellular concentrations at 2-3

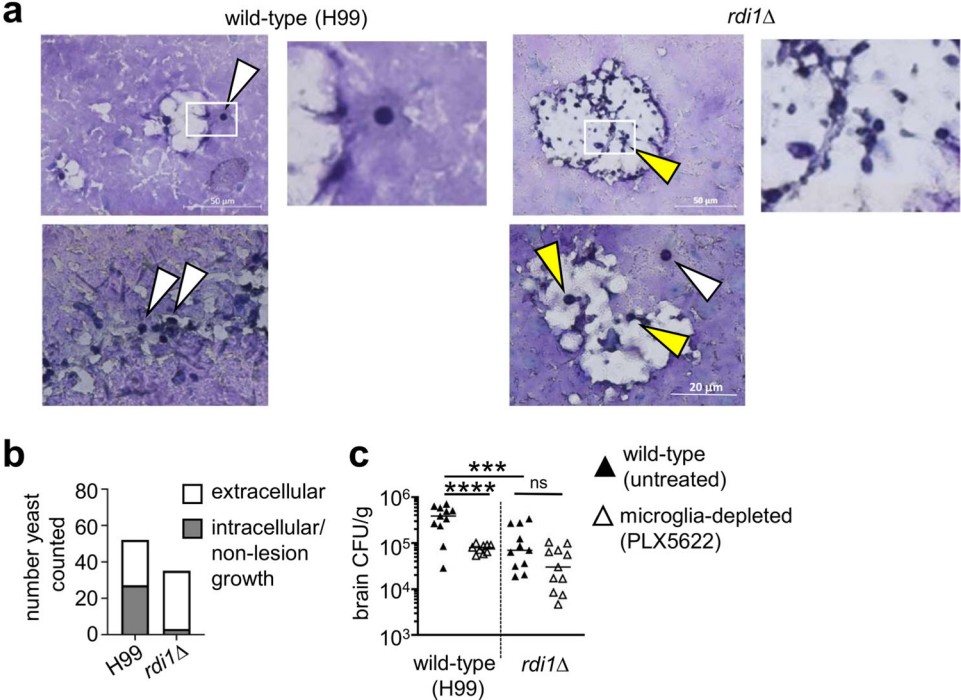

**Fig. 4 | Loss of microglia does not affect brain infection with C. neoformans rdi1Δ. a** Representative histology from brain of mice infected with wild-type *C. neoformans* H99 or *rdi1Δ C. neoformans* at day 3 post-infection, stained with PAS. White boxes in top images denote area of accompanying enlarged image. White arrows denote intracellular fungi, yellow areas denote extracellular fungi. Similar results were observed across 2 brains analysed. **b** Number of intracellular (grey bar) and extracellular (white bar) yeast counted across 3-5 sections from two mouse brains (for each strain). **c** Brain fungal burdens (median value denoted by bar) in untreated (black triangle) or PLX5622-treated (open triangle) mice at day 3 post-infection, infected with either wild-type *C. neoformans* H99 (*n* = 11 wild-type, *n* = 9 microglia-depleted) or *rdi1Δ C. neoformans* (*n* = 11 wild-type, *n* = 11 microglia-depleted). Data pooled from 2 independent experiments and analysed by one-way ANOVA with Bonferroni correction. ****$P < 0.0001$. ns = not significant. Source data are provided as a source data file.

orders of magnitude lower than intracellular concentrations[23]. We therefore tested the cellular localisation of fungal starvation responses in the brain, focusing on copper since *C. neoformans* response to copper starvation in the brain is a driver of virulence[24]. Under copper-starved conditions, *C. neoformans* upregulates expression of the copper importer *CTR4* (Fig. 5a), which is critical for fungal growth and virulence within the CNS[24]. We compared expression of *CTR4* by intracellular yeast cells (within microglia) to extracellular yeast in the brain in vivo, by measuring expression of a GFP transgene under the control of the *CTR4* promotor (*C. neoformans* pCTR4GFP) (Fig. 5b). We found that extracellular yeast had upregulated pCTR4GFP expression, in line with previous data that showed significant induction of this gene within the CNS[10] (Fig. 5c, d). In contrast, we found that yeast associated with microglia had significantly reduced pCTR4GFP expression (Fig. 5c, d). To ensure these results were not an artefact of different fluorescent properties of yeast inside host cells versus free yeast, we normalised pCTR4GFP expression to a constitutively-expressed mCherry housekeeper. Normalised pCTR4GFP expression was similarly reduced in intracellular yeast compared to extracellular yeast (Fig. 5e). Taken together, our data show that there is significant heterogeneity in *C. neoformans* populations in the brain, which may be driven by access to nutrients and trace metals such as copper.

### *C. neoformans* CTR4 expression occurs at < 50 nM copper

We next took advantage of the *pCTR4-GFP C. neoformans* strain to estimate copper concentrations in the CNS. We inoculated copper-deficient growth media with *C. neoformans pCTR4-GFP*, added increasing doses of copper to the media and compared GFP fluorescence after 24 hours growth by flow cytometry. At 10 μM copper, there was no detection of pCTR4GFP expression and yeast cells remained negative for GFP expression down to 100 nM copper (Fig. 5e). We began to detect expression of pCTR4GFP at 50 nM copper, where ~5% yeast cells were GFP+ (Fig. 5e). However, at 10 nM copper this rose significantly to ~90% GFP +, similar to the copper starved condition (Fig. 5e). These data indicate that *CTR4* expression switches on at copper concentrations <50 nM and does not exhibit a graded expression across a range of copper concentrations. Based on these data, we estimate that copper concentrations in the extracellular compartment of the *C. neoformans*-infected brain are between 10-50 nM, whereas copper concentrations within *C. neoformans*-infected microglia may be 50-100 nM or higher.

### *C. neoformans* undergoes copper starvation in brain macrophages

After confirming that microglia protected *C. neoformans* from copper starvation in the brain, we next analysed fungal expression of pCTR4GFP in other myeloid cell types to determine whether intracellular protection from copper starvation was unique to microglia. For that, we performed the same experiment as before using flow cytometry to determine the frequency of pCTR4GFP-positive yeast in the different cellular compartments. In contrast to yeast within microglia, we discovered that yeast internalised by macrophages in the brain had increased expression of pCTR4GFP, almost to the same levels as extracellular yeast (Fig. 6a). Intracellular fungi within recruited monocytes and neutrophils had variable levels of pCTR4GFP expression, which on average was intermediate between microglia and macrophages (Fig. 6a). Taken together, these data indicate that there is divergent access to copper between different myeloid cell types in the infected brain, resulting in variability in fungal starvation responses in this tissue.

### IFNγ induces copper restriction in fungal-infected microglia

IFNγ is a pro-inflammatory cytokine that is required for protective immunity against *C. neoformans* infection[25]. IFNγ mediates protective crosstalk between CD4 T-cells and macrophages, driving macrophage

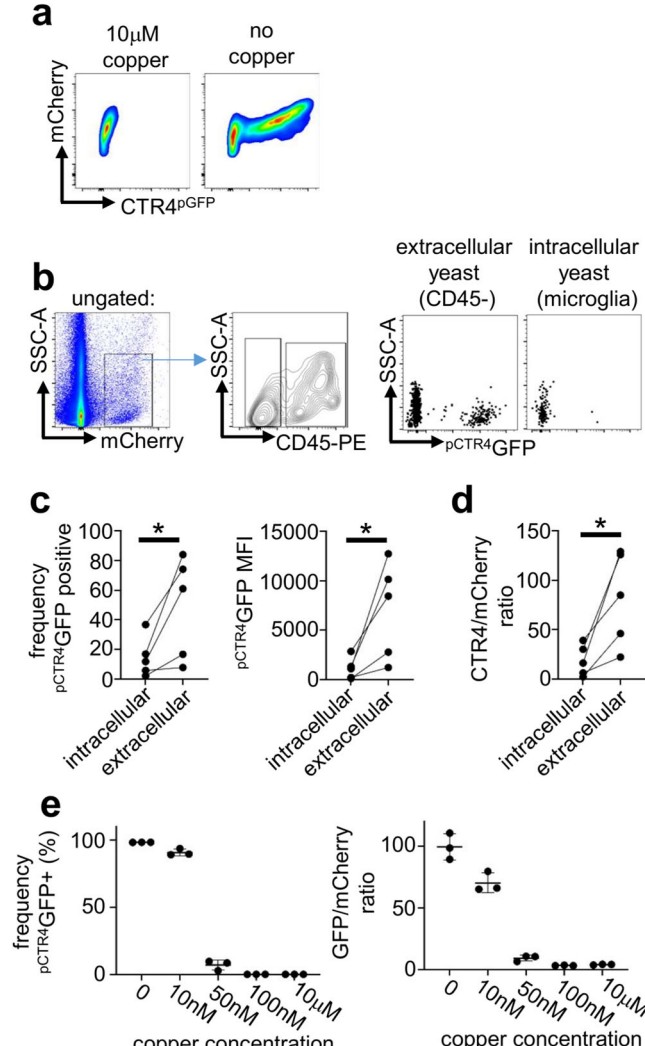

**Fig. 5 | C. neoformans is protected from copper starvation when associated with microglia. a** Example flow cytometry plots of pCTR4GFP *C. neoformans* grown in copper-deficient YNB media with and without copper supplementation for 18 hours. mCherry is under the control of the *ACT1* promotor and acts as a housekeeper, GFP expression is controlled by the *CTR4* promotor. **b** Gating strategy for analysing *pCTR4-GFP* yeast cells in the brains of mice at day 7 post-infection. Yeast were first gated on using the mCherry marker, then split into extracellular (CD45−) and intracellular (CD45+) groups. Intracellular yeast were further gated to specifically analyse microglia (Ly6G⁻Ly6C⁻CD45intCX3CR1hi). **c** Frequency (*P* = 0.0439) of GFP+ cells and median fluorescent intensity (MFI; *P* = 0.0379) of pCTR4GFP in yeast cells that were extracellular or intracellular within microglia. Each point represents an individual mouse. Data are pooled from two independent experiments and analysed by paired two-tailed t-test. *\**P* < 0.05. **d** Ratio (*P* = 0.0439) between GFP and mCherry in in yeast cells that were extracellular or intracellular within microglia. In panels c and d, each point represents an individual mouse. Data are pooled from two independent experiments and analysed by paired two-tailed t-test. *\**P* < 0.05. **e** Frequency of GFP+ cells and GFP/mCherry ratio in pCTR4GFP *C. neoformans* grown in copper-deficient YNB media supplemented with indicated concentrations of copper. Yeast cells were analysed by flow cytometry after 18 hours growth. Each point represents a technical replicate. Data are representative of 3 independent experiments and shown as mean +/- SEM. Source data are provided as a source data file.

activation via STAT1 signalling to produce nitric oxide and initiate fungal killing[26]. IFNγ has also been shown to modulate copper concentrations within macrophage phagosomes inducing either copper deprivation[27] or importing excess copper to induce toxicity[28]. We first explored whether differences in responses to IFNγ could partially

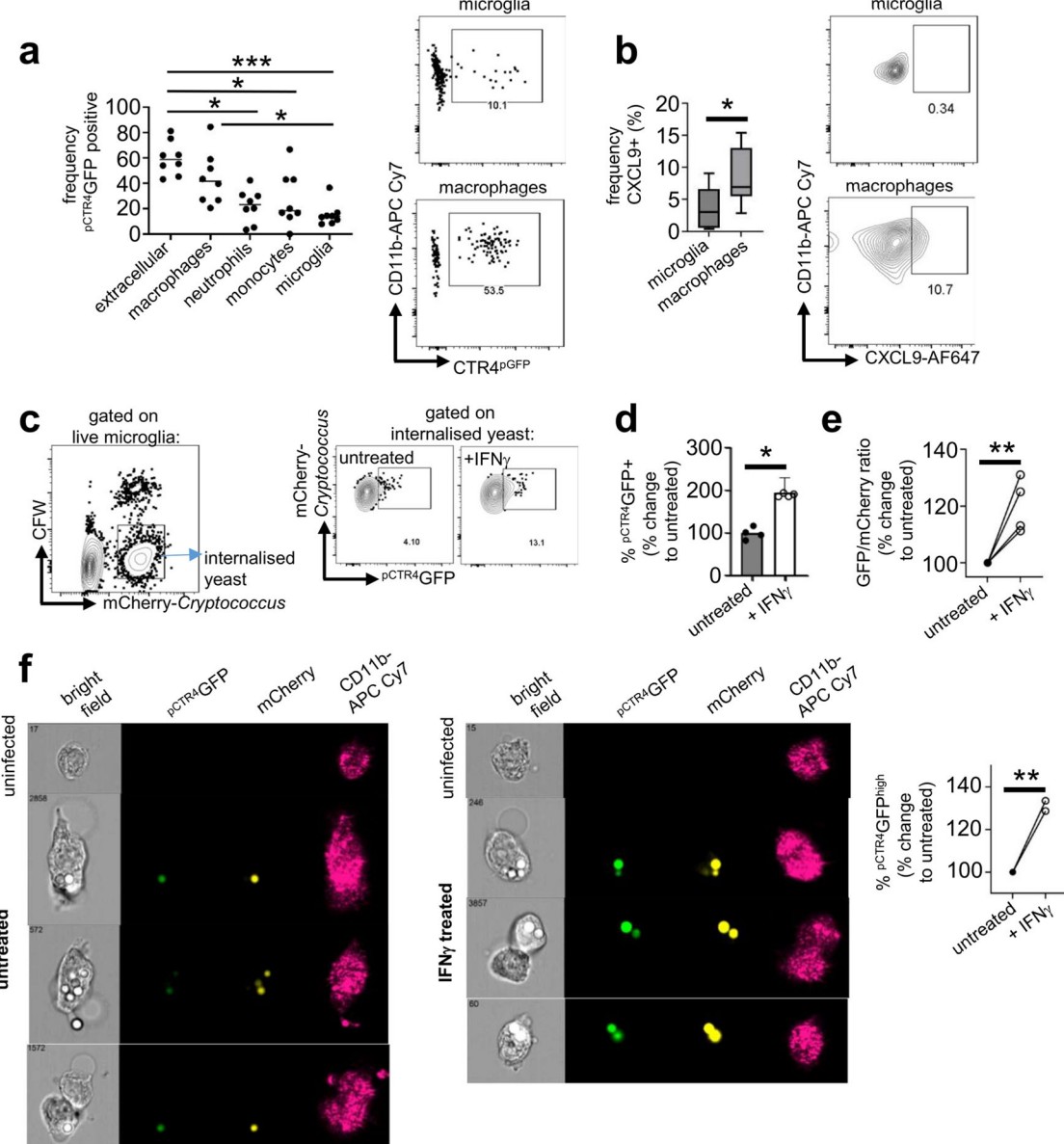

**Fig. 6 | Fungal heterogeneity in CTR4 expression correlates with IFNg stimulation of host cells. a** Expression of pCTR4GFP by yeast was assessed by flow cytometry in the indicated cell types as in other experiments. Each point represents an individual mouse analysed ($n = 8$ mice total). Data is pooled from three independent experiments and analysed by one-way ANOVA (***$P = 0.0003$, *$P = 0.043$ extracellular vs monocytes, *$P = 0.134$ extracellular vs neutrophils, *$P = 0.0327$ microglia vs macrophages). Representative FACS plots for microglia and macrophages within the same mouse are shown as an example. **b** Frequency of CXCL9+ microglia and macrophages in the infected brain at day 7 post-infection. Data pooled from three independent experiments ($n = 9$ mice) and analysed by unpaired two-tailed t-test. Box plots show median with 25%/75% percentiles and maximum and minimum values. *$P < 0.0172$. **c** Gating strategy used to compare pCTR4GFP expression by intracellular yeast within untreated or IFNγ-treated microglia. Cells were first gated to exclude free yeast, doublets and dead cells. Yeast bound to the surface of microglia but not internalised were removed from analysis using a calcfluor white (CFW) counter stain for the fungal

cell wall. **d** Frequency of intracellular pCTR4GFP+ yeast in untreated and IFNγ-treated microglia after 2 hours of infection. Bars represent mean data from 4 independent experiments, points show technical replicates from one representative experiment. Data are analysed by unpaired two-tailed t-test on pooled means ($P = 0.0266$). **e** Ratio between mCherry and pCTR4GFP expression by intracellular yeast in untreated and IFNγ-treated microglia after 2 hours of infection. Each point represents mean value from 2 or 3 technical replicates tested in 4 independent experiments. Data are analysed by unpaired two-tailed t-test ($P = 0.0055$). **$P < 0.01$. **f** BV2 cells were infected and stained as above prior to analysis with an imaging flow cytometer. Example images of untreated BV2 cells (left) and IFNγ-treated BV2 cells (right) are shown alongside their respected uninfected control (top row). Data was quantified by measuring the frequency of cells within the pCTR4GFPhigh gate, set within GFP + BV2 cells that were singlets and in focus. Data pooled from two independent experiments and analysed by two-tailed unpaired t-test ($P = 0.0059$). **$P < 0.01$. Source data are provided as a source data file.

explain the increased copper starvation response by yeast internalised within macrophages in contrast to microglia. For that, we compared microglia and macrophages in their production of CXCL9, a canonical chemokine regulated by IFNγ[29], as a marker for downstream IFNγ stimulation in the brain using intracellular flow cytometry. Those experiments showed that a significantly higher proportion of macrophages produced CXCL9 compared to microglia (Fig. 6b and Fig S4).

As this data indicated that copper starvation responses of intracellular fungi may correlate with IFNγ stimulation of host cells, we next examined whether IFNγ could modulate copper starvation responses

of fungi within microglia. We infected the murine microglia cell line BV-2 with *C. neoformans pCTR4-GFP*, comparing ^pCTR4^GFP expression by yeast internalised within untreated microglia, or microglia pre-treated with IFNγ (Fig. 6c). We found that yeast internalised by IFNγ-treated microglia had significantly higher expression of ^pCTR4^GFP than yeast internalised by untreated microglia (Fig. 6d, e). We repeated this experiment using an imaging flow cytometer and independently verified the increased expression and intensity of ^pCTR4^GFP by yeast internalised in IFNγ-treated microglia compared to untreated (Fig. 6f). These data suggest that IFNγ induces copper restriction within microglia and macrophage phagosomes resulting in an enhanced fungal copper starvation response.

### Tissue-resident microglia and infiltrating myeloid cells have differential abundance of trace metals

Our experiments with the ^pCTR4^GFP *C. neoformans* reporter strain indicated that intracellular residence within different myeloid cell types in the brain has a significant influence on fungal starvation responses. We therefore measured abundance of trace metals that the fungus may acquire from the host in myeloid cell subsets purified from the infected brain to better understand the drivers of this heterogeneity within the fungal population. We found that microglia had lower abundance of iron, magnesium and zinc than monocytes and macrophages, while neutrophils were somewhat variable (Fig. 7a). The limit of sensitivity in these experiments was too low to directly detect copper in purified cells, and hence we can only estimate access to copper within different cell types using the ^pCTR4^GFP reporter strain (as shown in Fig. 6a). Instead, we measured copper content in whole brain at various time points post-infection. Those experiments showed that

copper content remains relatively stable in the brain throughout infection and does not significantly alter from baseline (Fig. 7b), reflecting the tight homeostatic control on copper levels in tissues. Collectively, these data show that microglia do not appear to have greater abundance of trace metals than other cell types, indicating that differential access to these nutrients may be what drives divergent environmental responses by internalised fungi.

### Copper restriction by host myeloid cells does not determine fungal killing capacity

Since yeast within microglia and macrophages in the brain had different rates of ^pCTR4^GFP expression (Fig. 6a) and we measured differential abundance of other trace metals in these cell types, we next aimed to determine how host restriction of copper and/or access to other trace metals affected fungal viability and killing by different subpopulations of myeloid cells in the brain. We infected mice with GFP-expressing *C. neoformans* and used this to identify fungal-infected myeloid cells, as before (Fig. 1). We FACS purified fungal-infected cells and plated known numbers onto fungal growth media, counting the viable colonies that grew to determine rate of fungal killing and intracellular residence in each of the different cell types ex vivo. We found that neutrophils had a high rate of fungal killing and low intracellular residence, since only ~14% of yeast cells were viable from this cell type (Fig. 7c). In contrast, recruited monocytes had high intracellular fungal residence, with ~75% of yeast remaining viable (Fig. 7c). Microglia and macrophages had similar rates of fungal residence and killing capacity, with ~60% of yeast remaining viable in each of these cell types (Fig. 7c). Taken together, this data shows that neutrophils have high fungistatic activity in the brain,

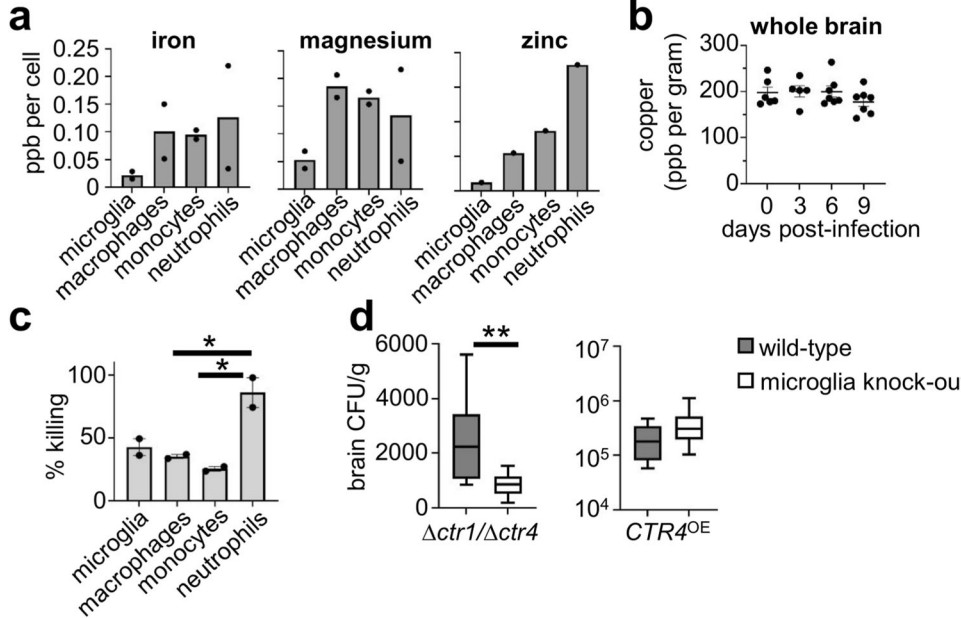

**Fig. 7 | *C. neoformans* CTR4 expression does not correlate with immune cell killing or metal abundance. a** Abundance of indicated metals in FACS-purified cell populations measured by ICP-MS. Total concentrations were divided by the number of cells used as input to generate the abundance (parts per billion, ppb) per cell. Data pooled from two independent sorts (iron, magnesium) or a single sort (zinc). **b** Total copper concentration in whole mouse brains at indicated time points post-infection. Each point represents an individual mouse ($n = 6$ mice at day 0, $n = 5$ mice at day 3, $n = 7$ mice at days 6 and 9). Data pooled from two independent experiments. **c** Ex vivo fungal killing assay. Mice were infected with GFP-expressing *C. neoformans* and GFP+ myeloid cells were FACS-purified at day 7 post-infection. Infected cells ($n = 1000$ cells in experiment 1, $n = 2000$ cells in experiment 2) were lysed in water prior to plating onto YPD agar plates. Colonies were counted to

calculate the percentage viability and frequency of fungal killing. Data pooled from two independent experiments, shown as mean +/- SEM and analysed by one-way ANOVA ($P = 0.0378$ macrophages vs neutrophils, $P = 0.0199$ monocytes vs neutrophils). *$P < 0.05$. **d** Brain fungal burdens at day 3 post-infection from *Sall1^CreER^Csf1r^flox^* mice infected intravenously with $5 \times 10^5$ *Δctr1/Δctr4 C. neoformans* ($n = 10$ wild-type mice, $n = 6$ microglia-depleted mice, $P = 0.0047$) or $2 \times 10^4$ CTR4^OE^ ($n = 5$ wild-type mice, $n = 8$ microglia depleted mice, $P = 0.2396$). Wild-type (grey bars) refers to Cre-negative littermate controls, microglia-depleted (open bars) refers to Cre+ littermates. Box plots show median with 25%/75% percentiles and maximum and minimum values. Data pooled from two independent experiments and analysed by two-tailed Mann Whitney U-test. **$P < 0.01$. Source data are provided as a source data file.

while macrophages, monocytes and microglia have similar rates of intracellular fungal growth.

### *C. neoformans* resistance to copper starvation overrides microglia-dependent fungal growth

Our data shows that intracellular fungi within microglia and macrophages have different expression of CTR4, indicating that there are divergent copper starvation responses between these intracellular infection niches. However, these responses did not correlate with fungal killing or abundance of trace metals. Myeloid cell killing may be impeded by fungal adaption to the intracellular environment, including upregulation of transporters to compensate for restricted access to metals such as copper. To better understand the relationship between *C. neoformans* copper starvation and fungal survival in the brain, we infected mice with *C. neoformans* mutant strains which are unable to compensate for copper restriction (Δctr1/Δctr4) or were insensitive to copper deficiency (CTR4^OE). The Δctr1/Δctr4 *C. neoformans* strain, which lacks the two major copper importers Ctr1 and Ctr4, has a significantly reduced capacity to infect the mouse brain (compare Fig. 7d with Fig. 2g), likely related to the growth defects of this strain due to the inability to import environmental copper. Importantly, in mice lacking microglia, growth of the Δctr1/Δctr4 mutant was even further restricted, indicating that expression of copper importers by *C. neoformans* and the presence of microglia are needed to mediate full virulence and infection in the brain (Fig. 7d). In contrast, a copper-insensitive strain of *C. neoformans* that overexpresses CTR4 was unaffected by the loss of microglia, and we recovered similar fungal burdens from wild-type and microglia-depleted mice (Fig. 7d). Taken together, this data shows that while microglia are not needed for protection against *C. neoformans* infection, they may influence the phenotype and survival of fungal cells in a manner that is partially dependent on the ability of *C. neoformans* to scavenge copper and adapt to a copper-restricted environment.

## Discussion

Microglia have been shown to play a central role in the protection against CNS infections, including *Candida albicans*[4], *Toxoplasma gondii*[3] and viruses[2]. Here, we show that microglia are largely redundant for the control of *C. neoformans* infection, and may support early *C. neoformans* growth in the brain of strains with high propensity to grow intracellularly within host myeloid cells. We found that microglia depletion reduces fungal brain infection at the early stages of infection, an effect that was dependent on intracellular growth of the fungus within microglia. While microglia did not contribute towards protection against infection, we found they did play a significant role in influencing the phenotype of fungal cells in the brain. Using a fungal reporter strain to track copper starvation responses, we show that the fungus is protected from copper starvation within microglia, leading to heterogeneity in fungal phenotype between different micro-niches in the brain.

Microglia can rapidly respond to microbial invasion of the CNS. For example, following *C. albicans* fungal brain infection, microglia produced IL-1β and CXCL1 within 24 hours of infection, which elicited a significant influx of neutrophils to the brain to enable clearance of the infection[4]. During infection with the parasite *T. gondii*, microglia rapidly produced IL-1α, which was required for the protective recruitment of inflammatory T-cells[3]. In our work, we found little evidence for a role of microglia in recruiting inflammatory myeloid cells to the CNS during *C. neoformans* infection, as we observed no consistent difference in the number of recruited neutrophils or monocytes to the infected brain of microglia-depleted mice in any of the models used. This lack of a role for microglia in early protective anti-cryptococcal immunity may be linked with poor induction of pro-inflammatory immune signalling in response to *C. neoformans*. Many fungi engage CARD9-coupled C-type lectin receptors on host immune cells, leading to a pro-inflammatory signalling cascade and leucocyte activation[5]. However, *C. neoformans* secretes a mannose-rich capsule which shields the fungal cell wall from recognition by CARD9-coupled receptors, such as Dectin-1[30]. Indeed, in contrast to other fungal infections, patients with deficiencies in CARD9 and related molecules do not appear to be more susceptible to cryptococcal meningitis[5], indicating that these pathways are redundant for protection against *C. neoformans* infection. The innate receptors mediating activation of brain-resident macrophages during *C. neoformans* infection remains to be determined, but this process likely depends on protective crosstalk with IFNγ-producing CD4 T-cells.

Deficiency or a functional defect in CD4 T-cells is the main risk factor for developing cryptococcal meningitis in humans[25]. Primary immunodeficiencies linked with cryptococcal meningitis primarily affect the crosstalk between CD4 T-cells and macrophages, such as loss-of-function mutations in *IL12RB1*[31] and *GATA2*[32]. In each of these cases, patients are thought to be susceptible to cryptococcal meningitis due to the lack of IFNγ-mediated activation of macrophages leading to poor fungal killing and intracellular fungal survival. In a model of chronic *C. neoformans* brain infection, microglia were activated after significant infiltration of pathologic IFNγ-producing CD4 T-cells which subsequently drove neuroinflammation and death[8]. In another study, treatment of rats with immunostimulatory immunotherapy caused microglial upregulation of MHCII and increased expression of several Toll-like receptors, which coincided with lymphocyte infiltration and fungal clearance[9]. These results indicate that microglia may be unable to respond to *C. neoformans* infection without T-cell help or immunotherapy. In our study, we have used an acute infection model and examined early time points (<1 week) prior to significant T-cell infiltration.

In addition to hosting intracellular fungi, microglia may also support infection by enabling fungal invasion across the blood-brain-barrier (BBB). *C. neoformans* is thought to invade the CNS using a variety of routes, including transcellular invasion of the BBB as free yeast or within host immune cells (e.g. monocytes)[33]. Whether microglia are involved with either of these processes is unclear. A small subset of microglia, called capillary-associated microglia (CAMs), have been shown to associate with blood vessels in the brain and are involved in maintaining integrity via purine signalling[34]. During sterile neuroinflammation, microglia migrate to cerebral blood vessels via CCR5 and drive claudin expression in endothelial cells[35], but during sustained or inappropriate inflammation microglia phagocytose components of the BBB and contribute to loss of integrity[35,36]. Loss of BBB integrity has been previously observed during cryptococcal meningitis, which is associated with fungal proliferation within cerebral blood vessels leading to rupture[37]. The role of microglia in BBB integrity during cryptococcal meningitis is still unresolved, and we cannot rule out that the reduction in brain fungal burdens observed in our microglia-depleted mice is partially due to effects on BBB integrity caused by depletion of CAMs and/or interruption of inflammatory microglia migration to the blood vessels. Indeed, we found that depletion of CX3CR1+ cells in the brain at late time points post-infection (which may include inflammatory border macrophages in addition to microglia) continued to reduce brain fungal burden, whereas this effect was absent in our other models. This may reflect differences in the loss of inflammatory subsets of brain macrophages and microglia that are specific to each model. Future studies should therefore examine microglia subsets and migration patterns during fungal infection to identify the different ways in which these cells contribute towards pathology.

Some of the microglia depletion methods used in our study, including the CSF1R inhibitor PLX5622[38,39], are reported to have off target effects on border macrophages within the CNS (e.g. perivascular macrophages). The role of non-microglia CNS border macrophages in the control and dissemination of *C. neoformans* is not fully understood,

but they have been shown to become heavily infected with the fungus and are localised to the main site of disease observed in humans[40]. It is possible that some of the depletion models used in our study may have interrupted fungal invasion tactics that involve these non-microglia populations, contributing to the reduction in fungal brain burdens we observed. To test this possibility, we generated *Sall1*<sup>CreER</sup>*Csf1r*<sup>flox</sup> animals which had specific depletion of microglia while leaving numbers of CNS border macrophages intact. In those animals, fungal burdens were also reduced in the brain, indicating that microglia are the main macrophages within the CNS that promote fungal brain infection at the early stages of infection. Further study is required to specifically examine the function of rarer border macrophages in cryptococcal meningitis and determine whether they contribute towards CNS dissemination and/or prevent fungal invasion across the BBB. For this, we require new models to deplete or manipulate the function of non-microglia macrophage populations in the CNS. Recent work by the Rua lab used an intracranial surgical model to apply PLX5622 directly onto the meninges to deplete macrophages at this border site while leaving brain-resident microglia intact[41]. This work revealed the critical role for these macrophages in preventing viral spread into other CNS tissues[41]. These novel approaches may prove a useful tool for future studies analysing fungal dissemination into the CNS and the role of the meninges in this process.

When we further examined the phenotype of intracellular yeast cells isolated from microglia, we discovered that the fungus was protected from copper starvation within these cells. Copper is an essential heavy metal critical for mitochondrial respiration and other biochemical processes[42]. *C. neoformans* strains lacking copper importers (and therefore intracellular copper) are severely restricted in their growth both in the host and laboratory media, which impacts on downstream regulation of growth and classic virulence traits such as iron uptake and melanin production[43]. Therefore, access to trace metals and micronutrients in different host niches could have a significant influence over fungal virulence and host responses. This concept has been demonstrated for the yeast *C. albicans*, where growth on different carbon sources affected exposure of immunostimulatory β-glucan within the cell wall which impacted on downstream host responses[44]. In our experiments, we discovered wide variation in the abundance of various elements in microglia and other myeloid cell types in the brain, indicating that the heterogeneity of yeast cells in the brain during infection may be vast. The impact of this on fungal responses to the microenvironment and consequences for pathogenesis will be an important future avenue to explore. It is important to note, however, that high abundance of trace metals may not necessarily reflect greater access by intracellular pathogens, as immune cells can restrict and traffic these elements to cellular compartments that do not contain yeast. It will therefore be critical to assess mammalian shuttling of metals and transporter expression in response to *C. neoformans* infection and/or cytokine stimulation in future studies.

In addition to the variation in metal abundance and infection rate that we observed across immune cells, it will be important to determine how diversity in *C. neoformans* isolates affect interaction with brain-resident myeloid cells. We found that infection with non-H99 strains of *C. neoformans* exhibited different phenotypes in microglia-deficient mice, although all strains showed no dependence on microglia for protection. Indeed, it has been found that there are significant differences across reference strains and clinical isolates in morphology (e.g. ability to form Titan cells)[18], melanin production[45] and antigenic differences in capsule formation[46]. All of these factors may affect interaction with host immune cells. Careful examination of how different *C. neoformans* isolates interact with tissue-resident myeloid cells and correlating this information with differences in expression of virulence factors will be an important next step to understand how the role of tissue-resident myeloid cells may be altered by fungal-specific factors.

In the CNS, intracellular copper levels are up to 1000-fold higher than extracellular levels[23]. This is likely because copper build-up within the CNS is highly toxic, leading to neuroinflammation of which inappropriate microglia activation is a hallmark[42]. For example, high copper concentrations have been measured in Alzheimer's plaques[47], and microglia surrounding those plaques have altered expression of copper transporters linked to pro-inflammatory cytokine responses[48]. Due to the restricted copper availability within the CNS, *C. neoformans* significantly upregulates copper importing proteins Ctr1 and Ctr4 within the brain[10,24]. Expression of these copper importers signifies fungal copper starvation, as they are not expressed when copper conditions are adequate and are required for growth in low copper[10]. Indeed, *C. neoformans* mutants deficient in *CTR4* have attenuated growth and virulence specifically within the CNS[10,24].

We tracked *C. neoformans* copper sensing using a *pCTR4-GFP* reporter strain. We found that *C. neoformans* highly upregulated CTR4 in the brain following intravenous infection, confirming previous work that showed *C. neoformans* is starved of copper in this tissue[10,24,49]. However, we found that CTR4 induction was significantly lower in yeast that were associated with microglia. These data indicate that copper concentrations and/or access to copper are higher within microglia. Similar observations have been made with the intracellular replicating fungus *Histoplasma capsulatum*. *H. capsulatum* uses the copper importer Ctr3 to protect against copper starvation[27], a protein that is functionally related to *C. neoformans* Ctr4. Infection of bone-marrow derived macrophages in vitro showed that *H. capsulatum* does not upregulate *CTR3* in the phagosome of resting macrophages and can acquire copper from this site, enabling its intracellular infection programme[27]. IFNγ stimulation of macrophages caused restriction of phagosomal copper concentrations, activating a copper-starvation response in *H. capsulatum* marked by upregulation of the copper importer *CTR3*[27]. In our study, we found evidence of differential responses to IFNγ between microglia and macrophages in the brain, which correlated with expression of CTR4 by intracellular fungi. These data indicate that pro-inflammatory cytokines may regulate access to copper by intracellular pathogens in a cell-type specific manner. However, we did not find differences in the ability of microglia and macrophages to restrict intracellular fungal growth ex vivo. It therefore remains unclear how IFNγ regulates anti-fungal immunity within the brain, and how restricting access to nutrients affects pathogenesis. Indeed, IFNγ was only found to boost fungal killing by macrophages when *H. capsulatum* lacked expression of CTR3, indicating that although IFNγ can drive nutritional immunity these mechanisms of protection can be subverted by fungal starvation responses that help fuel their intracellular growth and virulence programmes in nutrient restricted conditions. It will be important to discern the effects of IFNγ on different brain-resident myeloid cells during cryptococcal infection in future studies, including how receptor expression and signalling patterns differ between tissue-resident and recruited inflammatory myeloid cells. Adjunctive therapy with recombinant IFNγ in patients with HIV-associated cryptococcal meningitis decreased fungal burden in the CSF, a marker correlated with survival[50]. Yet, the specific protective mechanisms of IFNγ in anti-cryptococcal protection in the CNS are not fully resolved. Our data indicate that IFNγ may exert protective effects directly on microglia, at least partially, by limiting fungal access to key nutrients but the effectiveness of this may depend on fungal adaption to the specific microenvironment encountered within the cell type in question. Future studies will also need to examine other nutrients that the fungus may access within microglia, such as iron and lipids, which have both been shown to be acquired by *C. neoformans* within other macrophage types[51,52].

In summary, our data highlight microglia as an important reservoir for intracellular fungal infection, acting to shield the fungus from the copper-deficient environment within the CNS. This is one of the first examples of microglia acting to promote CNS infection, and

delivers new insights into critical pathways underlying host-fungal interactions in the brain during cryptococcal meningitis.

# Methods

## Ethics statement

All research complied with local ethical approval. Human tissue biopsies were collected and stored with consent for future research use under a study approved by the REC/Scientific Committee of the Infectious Disease Institute Makere University overseen by the Uganda National Council of Science and Technology. Animal studies were approved by the Animal Welfare and Ethical Review Board at the University of Birmingham and UK Home Office under Project Licence PBE275C33.

## Mice

8-12 week old mice (males and females) were housed in individually ventilated cages under specific pathogen free conditions at the Biomedical Services Unit at the University of Birmingham, and had access to standard chow and drinking water *ad libitum*. Mice were housed under 12 hour light/dark cycle at 20-24°C and 45-65% humidity. Experiments with transgenic mice utilised both males and females to maintain littermate controls. We used female mice for experiments with wild-type mice only, since female mice can be housed in larger groups/smaller cage numbers. Wild-type refers to C57BL/6NCrl (Charles River) or the corresponding littermates of genetically-modified lines; *Cx3cr1*-Cre[ER]*Rosa26*iDTR, *Sall1*-Cre[ER]*Rosa26*[Ai14], and *Sall1*-Cre[ER]*Csf1r*[flox]. *Cx3cr1*-Cre[ER], *Rosa26*iDTR, *Rosa26*[Ai14] and *Csf1r*[flox] mice were originally purchased from Jackson and colonies bred and maintained at the University of Birmingham. *Sall1*-Cre[ER] mice were a kind gift from Dr Melanie Greter (University of Zurich). Mice were euthanised by cervical dislocation at indicated analysis time-points, or when humane endpoints (e.g. 20% weight loss, hypothermia, meningitis) had been reached, whichever occurred earlier.

## Tamoxifen and diphtheria toxin treatments

Tamoxifen was dissolved in 100% ethanol (1 g/mL) and diluted in corn oil (Sigma) to 100 mg/mL. Mice received two doses of tamoxifen by oral gavage (10 mg in 100μL) 48 hours apart, prior to infection (experiments using Sall1-Cre[ER] crossed strains). In experiments analysing mice at day 6 post-infection, tamoxifen dosing was continued throughout infection with 48 hours between doses as before. In some experiments, mice were left to rest for 5-6 weeks prior to dosing with diphtheria toxin (experiments using Cx3cr1-Cre[ER] crossed strains). Diphtheria toxin (Sigma) was injected intraperitoneally (30 ng per gram body weight) daily for 3 days prior to infection, on the day of infection, and for 2 days after.

## PLX5622 treatment

PLX5622 (Plexxikon Inc. Berkley, CA) was formulated in AIN-76A rodent chow (Research Diets) at a concentration of 1200 mg/kg. Mice were provided with PLX6522 diet or AIN-76A control diet *ad libitum* for 1 week prior to infection, and continued throughout the infection study period.

## *C. neoformans* growth and mouse infections

*C. neoformans* strains used in this study were H99, KN99α-GFP[53], *rdi1Δ*[22], *ctr1Δ/ctr4Δ*[10], CTR4[OE 10], B3501[54], Zc15[21] and *pCTR4-GFP* (this study). Yeast was routinely grown in YPD broth (2% peptone [Fisher Scientific], 2% glucose [Fisher Scientific], and 1% yeast extract [Sigma]) at 30 °C for 24 hours at 200 rpm. The *ctr1Δ/ctr4Δ* mutant was grown in YPD broth supplemented with 10 μm copper sulphate (Fisher). In some experiments, *C. neoformans pCTR4-GFP* were grown in copper-deficient YNB media (Formedium), supplemented with 2% glucose and copper sulphate (Sigma; see figure legends for specific concentrations). For infections, yeast cells were washed twice in sterile PBS, counted using haemocytometer, and 2x10^4 yeast injected intravenously into the lateral tail vein. For analysis of brain and lung fungal burdens, animals were euthanized and organs weighed, homogenized in PBS, and serially diluted before plating onto YPD agar supplemented with Penicillin/Streptomycin (Invitrogen). Colonies were counted after incubation at 37 °C for 48 hours.

## Generation of pCTR4-GFP C. neoformans

The *pCTR4-GFP-mCherry* strain was generated by transforming the integrative *pCTR4-GFP* plasmid[10], which harbours GFP encoding gene driven by copper deficiency inducible promotor (*CTR4*), into KN99α-mCherry. The mCherry strain was a gift from Dr. Tongbao Liu (Southwest University, China), mCherry expression is controlled by the *ACT1* promotor. Transformations of *C. neoformans* were completed using biolistic transformation, as previously described[55]. The resulting *pCTR4-GFP-mCherry* strain was validated through growth on copper-sufficient and copper-deficient media to check correct functioning of the [pCTR4]GFP construct.

## Analysis of Brain Leukocytes by FACS

Leukocytes were isolated from brain using previously described methods. Briefly, brains were aseptically removed and stored in ice-cold FACS buffer (PBS + 0.5% BSA + 0.01% sodium azide) prior to smashing into a paste using a syringe plunger. The suspension was resuspended in 10 mL 30% Percoll (GE Healthcare), and underlaid with 1.5 mL of 70% Percoll. Gradients were centrifuged at 1000 g for 30 min at 4 °C with the brake off. Leukocytes at the interphase were collected and washed in FACS buffer prior to labelling with fluorophore-conjugated antibodies and flow cytometry analysis. In some experiments, cells were fixed and permeabilised using the eBioscience Foxp3 Staining Buffer Set prior to intracellular staining.

## Flow Cytometry

Isolated leukocytes were resuspended in PBS and stained with Live/Dead stain (Invitrogen) on ice as per manufacturer's instructions. Fc receptors were blocked with anti-CD16/32 and staining with fluorochrome-labelled antibodies (used at 1:200 final dilution) was performed on ice. Labelled samples were acquired immediately or fixed in 2% paraformaldehyde prior to acquisition. Anti-mouse antibodies used in this study were: CD45 (30-F11), CD11b (M1/70), CX3CR1 (SA011F11), MHC Class II (M5/114.15.2), F480 (BM8), Ly6G (1A8), Ly6C (HK1.4), CXCL9 (MIG-2F5.5) all from Biolegend. Samples were acquired on a BD LSR Fortessa equipped with BD FACSDiva v9.0 software. Analysis was performed using FlowJo (v10.6.1, TreeStar).

## Sorting of Brain Immune Cells and ex vivo Killing Assay

Mice were infected with GFP-expressing *C. neoformans* as above, and brain leukocytes isolated at day 7 post-infection. Infected (GFP+) microglia (CD45[int]CX3CR1[hi]CD11b[+]), macrophages (CD45[hi] CX3CR1[+]CD11b[+]F4/80[+]), monocytes (CD45[hi]CD11b[+]Ly6C[hi]Ly6G[-]) and neutrophils (CD45[hi]CD11b[+]Ly6G[+]Ly6C[int]) were FACS purified using a BD FACS Aria Fusion cytometer under sterile conditions. Collected cells (1000-2000 total) were centrifuged, the pellet resuspended in 1 mL sterile tissue-culture grade water, and samples were then plated onto YPD agar plates (200-300μL per plate). Plates were incubated at 30°C for 2 days prior to colony counting, and killing rate calculated as a percentage of colonies counted over total number of infected cells plated.

## Metal Measurements by Inductively-Coupled Plasma Mass Spectrometry (ICP-MS)

Whole brains were added to pre-weighed metal-free falcon tubes (Fisher) containing 1 mL 65% nitric acid diluted in metal-free analytical grade water (Fisher Chemicals). Brains were incubated at 85°C for two hours until dissolved, and then diluted to 8% nitric acid using metal-free water. FACS-purified cells were centrifuged and the pellet

resuspended in 100-300µL 65% nitric acid and samples incubated and diluted as above. All samples were analysed on a Perkin Elmer Nexion 300X Mass Spectrometer. Samples were introduced to the ICP at 0.3 mL/minute using a peristatic pump, with a helium flow of 5 mL/minute, and argon gas flow settings as follows: nebuliser 0.9 L/minute, auxiliary gas 1.2 L/minute, plasma gas 18 L/minute. The ICP was run in KED mode with Rf power set at 1600W.

## Tissue Culture

BV2 cells (a kind gift from Dr Michail Lionakis, NIH) were routinely maintained at 37 °C, 5% $CO_2$ in RPMI (supplemented with GlutaMax and HEPES, Gibco), further supplemented with 10% heat-inactivated foetal bovine serum (Gibco) and 1% Penicillin/Streptomycin (Invitrogen), and were split every 2-3 days after reaching 80-90% confluence. For experiments, BV2 cells were lifted using Trypsin-EDTA (Sigma) and a cell scraper, counted using Trypan blue exclusion, and seeded into 12 well plates, at $2 \times 10^5$ cells per well in 1 mL media. In some wells, media was further supplemented with 100 ng/mL recombinant mouse IFNγ (Biolegend). After 24 hours, BV2 cells were infected with $1 \times 10^6$ *C. neoformans* CTR4$^{pGFP}$ yeast that had been pre-opsonised with 10µg/mL anti-glucuronoxylomannan (GXM) antibody (18B7; Millipore) for 15 minutes at room temperature. After 2 hours, plates were placed on ice, media removed and replaced with 1 mL ice-cold 2 mM EDTA in 1xPBS. BV2 cells were lifted by gentle pipetting on ice and transferred to FACS tubes prior to staining with fluorophore-conjugated antibodies and 5µg/mL calcofluor white (Sigma), then acquired immediately on a BDFortessa as above.

## ImageStream

BV2 cells were seeded into 6-well plates at $1 \times 10^6$ cells per well in 2 mL, and cultured with or without IFNγ as outlined above. BV2 cells were infected at a ratio of 5:1 yeast to cells for two hours prior to lifting and staining as above. Cells were stained and washed in 1.5 mL Eppendorf tubes before resuspending in 100µL 2% paraformaldehyde. Fixed cells were run on an Amnis ImageStream X MkII equipped with three lasers and 6 optical detectors. Final analysis was completed using IDEAS software (version 6.2).

## Histopathology

Mouse brains were removed from infected mice at indicated time points and either fixed in 10% formalin for 24 hours before embedding in paraffin wax, or frozen in OCT before sectioning. Tissue sections were stained with periodic Acid-Schiff (PAS) and hematoxylin and eosin (H&E). Imaging of tissue sections were captured using a Zeiss Axio Slide Scanner and data analysed using Zen Lite Blue (version 1.1.2.0).

## Confocal Microscopy

Whole mouse brains were fixed in 4% paraformaldehyde for 24 hours at 4 °C then placed in 30% sucrose solution at 4 °C for 24 hours (or until the brain sank), before freezing in OCT (Epredia). Brains were sectioned horizontally (at 30 µm) using a cryostat (Bright Instruments). Slides were washed in ice-cold PBS prior to blocking for 1 hour at room temperature in blocking solution (0.1 M Tris, 1% BSA, 0.1% donkey serum, 0.05% Triton X-100, 0.01% Saponin). Iba-1 antibody (1:500, Wako) staining was performed in blocking solution overnight at 4 °C. Alexa-Fluor 647 secondary antibody (1:500, Thermofisher) was incubated for 2 hours at room temperature and nuclei were stained with DAPI (1:1000). Autofluorescence quenching was performed with the TrueVIEW Autofluorescence Quenching Kit (VectorShield) prior to mounting the slides wit ProLong Gold Antifade mountant (Thermofisher). The slides were imaged using the Zeiss Axio Scan Z1 slide scanner and the Zeiss LSM 880 Confocal with Airyscan Fast. Image analysis was performed using the Zen Blue (v3.1) software.

## Statistics

Statistical analyses were performed using GraphPad Prism 9.0 software. Details of individual tests are included in the figure legends. In general, data were tested for normal distribution by Kolmogorov-Smirnov normality test and analysed accordingly by unpaired two-tailed *t*-test or Mann Whitney *U*-test. In cases where multiple data sets were analyzed, two-way ANOVA was used with Bonferroni correction. In all cases, *P* values < 0.05 were considered significant.

## Reporting summary

Further information on research design is available in the Nature Portfolio Reporting Summary linked to this article.

## Data availability

All raw data associated with the figures is available on request, without restrictions. Source data are provided with this paper.

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

## Acknowledgements

We would like to thank the technical staff at the Biomedical Services Unit (Birmingham) for their care and help with animal husbandry. In particular, we thank Claire Lyons and Karen Boswell for their time and patience performing infections and caring for our animals. We thank Dr Guillaume Desanti and support staff at the Technology Hub and Flow Cytometry Unit at the University of Birmingham for their support with sorting, flow cytometry and microscopy experiments. This work was funded by the Academy of Medical Sciences (SBF004_1008, awarded to RAD), Medical Research Council (MR/S024611, awarded to RAD), Research Foundation Flanders (PhD studentship awarded to EV, 1SF2222N), the National Natural Science Foundation of China (31870140, awarded to CD) and the Wellcome Trust and the Royal Society (211241/Z/18/Z, awarded to ERB).

## Author contributions

S.H.M., M.S.F., S.H., A.A., E.V., Y.L., E.B. and R.A.D. performed the experimental studies. S.H.M., M.S.F., A.A., E.V., E.R.B., R.L., G.V.V. and R.A.D. carried out the analysis. ERB, RCM, CD, GVV and RAD supervised the work. R.L. performed the human pathology studies. R.A.D. obtained the funding. S.H.M., E.R.B., C.D., G.V.V. and R.A.D. conceived and designed experiments. S.H.M. and R.A.D. wrote the manuscript.

## Competing interests

The authors declare no competing interests.
