## [Peer Review File · Nature Communications]

Microglia are not protective against cryptococcal meningitisREVIEWER COMMENTS

Reviewer #1 (Remarks to the Author):

This exciting study provides novel insights regarding the role of microglia in cryptococcal infection. The authors studied the effect of microglia depletion on cryptococcal meningitis using a series of transgenic and chemical-based microglia depletion approaches. They showed that microglia become a host for intracellular fungal growth. The data all line up in support of microglia becoming promiscuous to the fungal growth by protecting fungus from copper starvation and serving as a site of microbial nutrient acquisition, which is very interesting. This study also expands and extends previous findings (Hazra et al 2018, Scand. J. Imm.) that proinflammatory immunotherapy improves microglial responses to cryptococcus, demonstrating that IFN-g therapy impacted starvation responses by intracellular fungi and suggesting copper restriction in microglia could be one of the mechanisms that limit microglia hijacking by *C. neoformans*.

The strengths of the manuscript are the use of multiple models to demonstrate the role of microglia and the use of novel cryptococcal mutants to analyze the role of copper availability/deprivation.

There are also some limitations, such as the use of only one cryptococcal strain genetic background to demonstrate the effect of microglia depletion, that some readouts are indirect, and that the effect on fungal growth in the brain was studied in an extremely short time point post-infection, which typically last for several weeks.

Major comments.

1) Virtually all comparisons of fungal burden are from day 3, with the exception of day 5 shown for one other model.

Other data showing the effect of microglia depletion reach as far as day 7, but not the fungal burdens. Was the beneficial effect of microglia depletion restricted to the very early timepoint? If the effect wanes over time is it linked to the arrival of T-cells into the brain or with the expression of some cytokines?

2) While strain H99 is the most commonly published cryptococcal strain, it is also one showing quite extreme virulence level and capability to hijack mononuclear phagocytes. Demonstration that other strains of cryptococcus can benefit from the presence of microglia would help to assess the biological relevance, especially if a less virulent strain could show an even more drastic phenotype.

3) Figures 3 show that microglia are the preferred intracellular infected cell subset and Fig 5 that the "copper-sensing" mutant is no longer copper starved when associated with microglia. Is the latter unique to microglia and how this compares to monocytes which are also quite numerous but harbor less fungus in the CNS? Are monocytes and microglia different in terms of the uptake, harboring, and killing of the fungus in vitro and is this related to copper content?

4) The copper content in the environment is estimated based on pCTR4-GFP expression by the fungus and while elegant is not a direct measurement. Can a more direct approach be used such as one in PNAS 102 (32) 11179-11184 be used to define copper concentration in microglia and other cells?

Also in Fig 6, it is unclear, whether IFN-g stimulation increasing the fungistatic activity of microglia depends directly on copper. Can an increasing copper concentration strategy be used in IFN-g-activated microglia to clarify this point?

Minor comments

1) Since microglia have different effects on different fungi in brain infections the title should probably be restricted to *C. neoformans*

2) Quality of histology Fig 4a can be increased by increasing magnification and adding arrows to point out the *C. neoformans*.

3) Study by Hazra et al 2018, Scand. J. Imm. should be cited and discussed in the context of this work

Reviewer #2 (Remarks to the Author):

This manuscript described the role of microglia during *Cryptococcus neoformans* infection in the brain. The study showed that microglia do not protect mice from *Cryptococcus* brain infection. Instead, microglia provided a niche for the fungus to survive within the cell. The author also tried to connect copper and yeasts in microglia in the last half.

The strengths of this study include the intriguing finding that microglia do not protect mice from *Cryptococcus* infection. This is interesting because myeloid cells (including microglia) are generally considered to be antifungal. Yet, the result of microglia providing a place to stay for *Cryptococcus* may be reasonable, considering that peripheral myeloid cells carry fungi to the brain by the Trojan horse approach. The authors used multiple mouse systems to deplete microglia, and the approach is solid and highly appreciated. However, the last half of the copper-mediated approach is not strong. Weaknesses in the copper part include confusing logic and not convincing data for IFN γ , as described below. Thus, the first half of this manuscript is strong but the last half of the manuscript reduces the enthusiasm.

Details are described below.

The results of copper and pCTR4-GFP are confusing. Fig. 5a demonstrated that copper inhibits pCTR4-GFP expression, which is considered a *Cryptococcus* virulence factor. Fig. 5b and 5c showed no pCTR4-GFP expression in the fungus internalized by microglia. However, if microglia contribute to the virulence of *Cryptococcus*, does the fungus want to express pCTR4-GFP?

It is not clear if the increase of pCTR4-GFP yeasts by IFN γ is biologically significant due to a minute increase in GFP -- Fig. 6a right panels showed the increase of pCTR4-GFP+ yeasts, but the main population (pCTR4-GFP- yeasts) only faintly increased pCTR4-GFP intensity, which is shown to slightly cross the gating line.

Fig. 3d and 3e require definitive images with microglia internalizing yeasts.

Fig. 4 legend subtitle "Reduction of fungal brain burden in microglia-depleted mice depends on intracellular fungal growth" is confusing. Microglia provide a space for intracellular fungal growth based on the results, but the subject of the title sentence is the reduction of fungal burden in "microglia-depleted" mice?

Line 170 "Histological examination ... with the *rdi1* Δ mutant revealed primarily extracellular yeast in areas of tissue damage ..., whereas wild-type yeast cells were mostly intracellular at this time (Fig 4a)." Fig. 4a is not convincing for this description. Clear images and statistical analyses are required.

Additional comment (encouraged by the journal): Sex considerations for mice are not discussed (It only mentions that male and female are used).

Reviewer #3 (Remarks to the Author):

GENERAL COMMENT

For sure an interesting study by Mohamed et al providing important novel findings in the field of neuro-infections and neuroinflammation. Firstly, authors should introduce and explain what the classes of CNS-resident macrophages in the brain are. Inflammatory macrophages are not known as CNS-resident because those are usually the ones infiltrating into the brain upon neuroinflammation (from periphery). Certain dogmas, such as the primary role of microglia in fighting brain infections, can be reversed with the results provided by the authors. On the other hand, the study has important gaps in scientific concepts and lacks important experimental controls for the in vivo experiments and important in vitro experiments with primary mouse brain cells that should be performed in order to support the preliminary conclusions drawn by the authors. In addition, authors should provide more comprehensive evidence for what concerns the phagocytic capacity of both microglia and other CNS-resident macrophages towards C.

neoformans. The simple observation of microbes inside immune cells is not enough to conclude anything concerning the survival capacity of microbes inside immune cells. Lastly, the dataset concerning copper starvation are convincing for microglia, but authors have not investigated also other CNS-resident macrophages can retain copper intracellularly which can be used by microbes to survive and grow. Please, find below my comments, remarks and suggestions.

INTRODUCTION

Especially in the introduction, authors have the tendency to use the terms "CNS" and "brain" as synonyms, which is wrong.

RESULTS

The first title section is misleading. In what sense brain macrophages "support" fungal brain infection? The verb "support" is not appropriate.

Fig. 1 shows that the depletion of brain-resident macrophages is beneficial for the brain in reducing fungal infection. This finding is absolutely novel and important in the field, since the absence of the primary line of defense in the brain would be considered to be essential for fighting infections. Results of Fig. 1 should be supported by in vitro results using microglia, ideally primary microglia isolated from the same mice used, to assess the phagocytic capacity of microglia towards *C. neoformans* infection. If absence of microglia helps the brain in clearing the infection, it can be assumed that microglia don't have a strong phagocytic capacity towards fungal infection.

In the second section of the results, authors state on line 110 "CNS-resident macrophages, including inflammatory macrophages...". This concept is not clear. Upon neuroinflammation microglia are activated and trigger a neuroinflammatory process which recruits "for help" peripheral macrophages. These immune cells are not CNS-resident. Authors should clarify this concept. In addition, the experimental design is confusing. In the first section of the results, authors have performed CX3CR1hiCD45int microglia depletion, and such microglia depletion led to a significantly reduced brain fungal burden. In the second section of the results, authors wanted to perform "specific microglia depletion while leaving other CNS-resident macrophages intact". Authors should clarify the class of macrophages (CNS-resident) present in the brain, and what is their role and % of presence in the brain, assuming that microglia are known to be the primary resident brain macrophages.

In Figure 2 authors assessed that the absence of microglia, but normal numbers of meningeal macrophages and other macrophage populations, led to significantly reduced brain fungal burden. An important control would be to have mice with depleted meningeal macrophages and other CNS-resident macrophages but normal number of microglia and assess if brain fungal burden is not reduced. Alternatively, as suggested above, microglia and other CNS-resident macrophages should be isolated from mouse brains and phagocytosis capacity towards fungal infection should be analyzed in vitro. Comparison of the phagocytic capacity of microglia and CNS-resident macrophages can provide an explanation of the scenario observed in vivo using Sall1CreERCsf1rflox mice.

The third section of results set a very important new concept in the neuro-infection field, which is that microglia are not the primary immune cells triggering the neuroinflammatory response that leads to the recruitment of peripheral immune cells into the brain. If this concept is true, or partially true, authors should also use mice with depleted CNS-resident macrophages and intact microglia. Alternatively, primary microglia and CNS-resident macrophages should be isolated from mouse brains and release of neuroinflammatory cytokines and chemokines should be analyzed upon infection in vitro with *C. neoformans*. Such experiment would provide a proof that microglia or other CNS-resident macrophages are involved, or not, in the neuroinflammatory response during fungal infections.

The analysis of intracellular fungal residence within microglia is not accurate. The simple observation of microbes inside microglia or macrophages is not an absolute indication of either microbial survival or killing by the immune cell. Visualization of pathogens inside immune cell should be accompanied by a co-staining of an autophagolysosomal marker, such as LC3B, to have an indication if the intracellular microbes are degraded or undergoing a degradation path. If such co-localization is not detected, then it is justifiable to conclude that intracellular microbes are viable inside immune cells. Only the detection of microcolonies (more frequent for bacteria) inside immune cells is an indication of microbes growing inside immune cells. Alternatively, to assess in a quantification manner the capacity of microbes to grow inside immune cells, a phagocytosis (also known as gentamicin-protection assay, for bacterial infections, the same can be performed for fungal infections using an anti-fungal agent) should be performed in vitro. After gentamicin/anti-fungal treatment, all extracellular microbes are killed and immune cells with intracellular microbes

can be lysed at different time-points and plated. If the count of microbial colonies is increasing over time, it means that microbes are growing inside immune cells. Authors have focused only on the capacity of intracellular fungi to survive and grow inside microglia, however they have not considered to assess, on the other hand, the phagocytic capacity of microglia. Can microglia and CNS-resident macrophages overexpress phagocytic markers (like TGFbeta) upon fungal infection? Authors have shown that *C. neoformans* is protected from copper starvation within microglia. The results shown are convincing, however, this result section seems to point out that presence of microglia is helping *C. neoformans* to survive and grow well in the brain thanks to the good levels of copper inside microglia. This can be a valid explanation of the reduced brain fungal burden when microglia are absent. Authors should assess whether *C. neoformans* is protected from copper starvation also in other CNS-resident macrophages, using a mouse model in which microglia are depleted and CNS-resident macrophage levels are not altered. Is this scenario specific for microglia only? Authors have not excluded that *C. neoformans* is protected from copper starvation also in other CNS-resident macrophages.

In the last result section, authors state that "IFNgamma also has been shown to modulate copper concentrations within macrophage phagosomes...". However, in this result section, authors focus only on microglia without considering other CNS-resident macrophages. Also, authors use the term "phagosome", and the concept of phagocytosis usually implies that phagocytosed particles (microbes too) are eliminated by immune cells. Two are the unclear aspects of this result section: 1. Why authors have not analyzed CNS-resident macrophages too for what concerns the copper restriction induced by IFN gamma 2. Could the copper levels inside microglia/macrophages a way used by the immune cells to attract *C. neoformans* and phagocytose fungi? It's important to be not only in the "mind" of the microbes, but also the host cell perspective should be considered.

METHODS

How have authors assessed the levels of microglia after cell depletion post tamoxifen treatment without any infection prior to performing the experiments shown in the manuscript? Has depletion of microglia been always consistent?

Why have authors taken in consideration meningeal macrophages? Infection of the mice was done intravenously, so fungi have entered the brain via the blood through the BBB. Meningeal macrophages can be important in case of an intracisternal route of infection, but not primarily in this case. Have authors performed staining to assess by microscopy if meningeal macrophages are activated? Furthermore, meninges are extremely difficult to collect from mice. Why haven't authors considered pericytes, the perivascular macrophages of the BBB, which should definitely have a more important role in this study since the infection is blood-borne.

Reviewer #1 (Remarks to the Author):

*This exciting study provides novel insights regarding the role of microglia in cryptococcal infection. The authors studied the effect of microglia depletion on cryptococcal meningitis using a series of transgenic and chemical-based microglia depletion approaches. They showed that microglia become a host for intracellular fungal growth. The data all line up in support of microglia becoming promiscuous to the fungal growth by protecting fungus from copper starvation and serving as a site of microbial nutrient acquisition, which is very interesting. This study also expands and extends previous findings (Hazra et al 2018, Scand. J. Imm.) that proinflammatory immunotherapy improves microglial responses to cryptococcus, demonstrating that IFN-g therapy impacted starvation responses by intracellular fungi and suggesting copper restriction in microglia could be one of the mechanisms that limit microglia hijacking by *C. neoformans*.*

The strengths of the manuscript are the use of multiple models to demonstrate the role of microglia and the use of novel cryptococcal mutants to analyze the role of copper availability/deprivation. There are also some limitations, such as the use of only one cryptococcal strain genetic background to demonstrate the effect of microglia depletion, that some readouts are indirect, and that the effect on fungal growth in the brain was studied in an extremely short time point post-infection, which typically last for several weeks.

Major comment 1: *Virtually all comparisons of fungal burden are from day 3, with the exception of day 5 shown for one other model. Other data showing the effect of microglia depletion reach as far as day 7, but not the fungal burdens. Was the beneficial effect of microglia depletion restricted to the very early timepoint? If the effect wanes over time is it linked to the arrival of T-cells into the brain or with the expression of some cytokines?*

Response: In our study, we have used an acute infection model where animals do not typically survive beyond day 9 post-infection. We now include data examining the phenotype of microglia-deficient mice at day 6 post-infection using two of other models. We found that mice treated with PLX5622 had lost the phenotype at day 6 post-infection. In our Sall1-CreER-Csf1rflox model, microglia-deficient mice had a non-significant trend of less fungal brain burdens at day 6 post-infection. This new data is included in new Fig 4e. We believe the loss of phenotype late post-infection is linked to the increasing ratio of extracellular growth by yeast in the brain as these later time points. We include new data that quantifies extracellular growth in histology sections at day 3 and 6 post-infection (see new Fig 4d). This new data and its implications are discussed in the revised manuscript. In light of this new data, we have also updated the paper title to reflect microglia role in promoting early infection.

Major comment 2: *While strain H99 is the most commonly published cryptococcal strain, it is also one showing quite extreme virulence level and capability to hijack mononuclear phagocytes. Demonstration that other strains of cryptococcus can benefit from the presence of microglia would help to assess the biological relevance, especially if a less virulent strain could show an even more drastic phenotype.*

Response: We include new data in new Fig S3 that examines the phenotype of microglia-deficient mice when infected with *C. neoformans* recent clinical isolate Zc15 (VNB-A), and B3501 (var. *neoformans*). Both of these strains have reduced virulence compared to H99. Interestingly, microglia-deficient mice have similar fungal burdens when infected with these strains demonstrating how strain variation can alter host immune responses. We suspect that these strains may have reduced virulence because they are less capable of hijacking microglia than H99, thus the role of microglia may be somewhat redundant with these strains. We have added these points in the revised discussion as well as new citations about strain variation in *C. neoformans* and the impact on interaction with the host.

Major comment 3: *Figures 3 show that microglia are the preferred intracellular infected cell subset and Fig 5 that the "copper-sensing" mutant is no longer copper starved when associated with microglia. Is the latter unique to microglia and how this compares to monocytes which are also quite numerous but harbor less fungus in the CNS? Are monocytes and microglia different in terms of the uptake, harboring, and killing of the fungus in vitro and is this related to copper content?*

Response: We have now performed experiments comparing the copper-starvation response of the fungus within different host cell types in the brain including macrophages, monocytes and neutrophils (see new Fig 6a). We have also compared intracellular survival of the fungus in these different cell types ex vivo (new Fig 6c), and measured abundance of various trace metals in sort-purified cells from the infected brain (see new Fig 6b and response to next comment). Collectively these experiments show that the fungus undergoes copper starvation inside macrophages but not other myeloid cell types, and that there is differential abundance of metals between cell types. This does not appear to be related to killing capacity or ability to restrict intracellular survival of the fungus however, because we found little difference in these parameters between microglia, macrophages and monocytes. This reveals outstanding questions for the field for how nutritional immunity may benefit the host and the mechanisms used by brain myeloid cells to restrict fungal infection, points which we have added into the revised discussion.

Major comment 4: *The copper content in the environment is estimated based on pCTR4-GFP expression by the fungus and while elegant is not a direct measurement. Can a more direct approach be used such as one in PNAS 102 (32) 11179-11184 be used to define copper concentration in microglia and other cells? Also in Fig 6, it is unclear, whether IFN-g stimulation increasing the fungistatic activity of microglia depends directly on copper. Can an increasing copper concentration strategy be used in IFN-g-activated microglia to clarify this point?*

Response: We now include direct copper measurements in the whole brain at various time points post-infection (see new Fig 6). We attempted to measure copper content of FACS purified cell populations but the limit of sensitivity was not high enough. We did, however, successfully measure other elements that the fungus may acquire from the host, and instead present this new data (see new Fig 6). This shows that abundance of elements such as iron, magnesium and zinc are different between myeloid cell types in the infected brain which may drive divergent starvation/nutrient acquisition programs by the fungus in these different cell types. This data also shows that microglia do not appear to be specifically abundant in these metals, but rather that they may require outside help to restrict access to these nutrients from intracellular pathogens. We make all of these points in the revised manuscript.

Regrettably, we cannot perform the copper concentration strategy as proposed here, because the addition of exogenous copper to in vitro cultures is toxic to cells. Our data so far point to access (rather than abundance) mediating fungal starvation responses in microglia.

Minor comments

1) *Since microglia have different effects on different fungi in brain infections the title should probably be restricted to C. neoformans*

Response: We appreciate the suggestion by the reviewer, but felt that ‘fungi’ was more accessible to a wider readership as others outside the medical mycology field may not recognise *Cryptococcus* as a fungus. We will therefore defer to the editors for their guidance on the preferred title.

2) *Quality of histology Fig 4a can be increased by increasing magnification and adding arrows to point out the C. neoformans.*

Response: We have added increased magnification panels, additional images and arrows to Fig 4 as the reviewer suggested, as well as new quantification data from these images (see revised Fig 4).

3) *Study by Hazra et al 2018, Scand. J. Imm. should be cited and discussed in the context of this work*

Response: The suggested citation has been added and discussed in the revised manuscript.

Reviewer #2 (Remarks to the Author):

*This manuscript described the role of microglia during *Cryptococcus neoformans* infection in the brain. The study showed that microglia do not protect mice from *Cryptococcus* brain infection. Instead, microglia*

provided a niche for the fungus to survive within the cell. The author also tried to connect copper and yeasts in microglia in the last half.

The strengths of this study include the intriguing finding that microglia do not protect mice from Cryptococcus infection. This is interesting because myeloid cells (including microglia) are generally considered to be antifungal. Yet, the result of microglia providing a place to stay for Cryptococcus may be reasonable, considering that peripheral myeloid cells carry fungi to the brain by the Trojan horse approach. The authors used multiple mouse systems to deplete microglia, and the approach is solid and highly appreciated. However, the last half of the copper-mediated approach is not strong. Weaknesses in the copper part include confusing logic and not convincing data for IFN γ , as described below. Thus, the first half of this manuscript is strong but the last half of the manuscript reduces the enthusiasm.

Major comment 1: *The results of copper and pCTR4-GFP are confusing. Fig. 5a demonstrated that copper inhibits pCTR4-GFP expression, which is considered a Cryptococcus virulence factor. Fig. 5b and 5c showed no pCTR4-GFP expression in the fungus internalized by microglia. However, if microglia contribute to the virulence of Cryptococcus, does the fungus want to express pCTR4-GFP?*

Response: CTR4 is not a virulence factor *per se*, but rather CTR4 enables the fungus to acquire copper in conditions of limited availability. Copper is needed for general growth (not just in an infection setting) and helps fuel the expression of classic virulence traits (e.g. melanin production, iron uptake) through its role in respiration and cell growth. We now make this point clearer in our revised discussion. In addition, we have added new data where we infected microglia-deficient mice with a Ctr1/4-doubleKO strain of the fungus (see Fig 5b). This strain is avirulent but is able to establish brain infection when given as a high dose. However, in mice lacking microglia, the ability of this strain to infect the brain is even further reduced, demonstrating that the fungus can survive in copper-starved conditions when it either expresses the copper transporters CTR1/4 or has access to microglia. When both are removed, infection is severely compromised.

Major comment 2: *It is not clear if the increase of pCTR4-GFP yeasts by IFN γ is biologically significant due to a minute increase in GFP -- Fig. 6a right panels showed the increase of pCTR4-GFP+ yeasts, but the main population (pCTR4-GFP- yeasts) only faintly increased pCTR4-GFP intensity, which is shown to slightly cross the gating line.*

Response: We have now repeated this experiment using an imaging flow cytometer, which independently verified this result. We have included these results and representative images in new Fig 7.

Major comment 3: *Fig. 3d and 3e require definitive images with microglia internalizing yeasts.*

Response: We include new confocal images of intracellular yeast within microglia (see revised Fig 3).

Major comment 4: *Fig. 4 legend subtitle "Reduction of fungal brain burden in microglia-depleted mice depends on intracellular fungal growth" is confusing. Microglia provide a space for intracellular fungal growth based on the results, but the subject of the title sentence is the reduction of fungal burden in "microglia-depleted" mice?*

Response: We have reworded this figure title to more accurately reflect the data presented as per the reviewer's suggestion.

Major comment 5: *Line 170 "Histological examination ... with the rdi1 Δ mutant revealed primarily extracellular yeast in areas of tissue damage ..., whereas wild-type yeast cells were mostly intracellular at this time (Fig 4a)." Fig. 4a is not convincing for this description. Clear images and statistical analyses are required.*

Response: We now provide additional images, increased magnification of these histology images and quantification from these sections (see Fig 4 and response to reviewer 1, above).

Major comment 6: *Additional comment (encouraged by the journal): Sex considerations for mice are not discussed (It only mentions that male and female are used).*

Response: We have updated our methods section to include a sentence on sex considerations for our animal experiments.

Reviewer #3 (Remarks to the Author):

*For sure an interesting study by Mohamed et al providing important novel findings in the field of neuro-infections and neuroinflammation. Firstly, authors should introduce and explain what the classes of CNS-resident macrophages in the brain are. Inflammatory macrophages are not known as CNS-resident because those are usually the ones infiltrating into the brain upon neuroinflammation (from periphery). Certain dogmas, such as the primary role of microglia in fighting brain infections, can be reversed with the results provided by the authors. On the other hand, the study has important gaps in scientific concepts and lacks important experimental controls for the in vivo experiments and important in vitro experiments with primary mouse brain cells that should be performed in order to support the preliminary conclusions drawn by the authors. In addition, authors should provide more comprehensive evidence for what concerns the phagocytic capacity of both microglia and other CNS-resident macrophages towards *C. neoformans*. The simple observation of microbes inside immune cells is not enough to conclude anything concerning the survival capacity of microbes inside immune cells. Lastly, the dataset concerning copper starvation are convincing for microglia, but authors have not investigated also other CNS-resident macrophages can retain copper intracellularly which can be used by microbes to survive and grow. Please, find below my comments, remarks and suggestions.*

Major comment 1: *Especially in the introduction, authors have the tendency to use the terms "CNS" and "brain" as synonyms, which is wrong.*

Response: We have updated the introduction to make clear when we mean brain specifically and corrected throughout.

Major comment 2: *The first title section is misleading. In what sense brain macrophages "support" fungal brain infection? The verb "support" is not appropriate.*

Response: We have changed the word 'support' to 'increase' in the subsection title.

Major comment 3: *Fig. 1 shows that the depletion of brain-resident macrophages is beneficial for the brain in reducing fungal infection. This finding is absolutely novel and important in the field, since the absence of the primary line of defense in the brain would be considered to be essential for fighting infections. Results of Fig. 1 should be supported by in vitro results using microglia, ideally primary microglia isolated from the same mice used, to assess the phagocytic capacity of microglia towards *C. neoformans* infection. If absence of microglia helps the brain in clearing the infection, it can be assumed that microglia don't have a strong phagocytic capacity towards fungal infection.*

Response: We include new data in the revised manuscript that compares intracellular survival of the fungus in primary microglia, macrophages, monocytes and neutrophils isolated ex vivo from infected mice (see new Fig 6). These data, coupled with our fungal-tracking experiments presented in Fig 3, demonstrate that microglia have a poor uptake and killing rate of the fungus compared to professional antifungal cells such as neutrophils. This data is aligned with other published studies that demonstrate the poor phagocytic capacity of microglia compared to other tissue-resident macrophages, and that phagocytosis of this fungus is hampered by its capsule which reduces uptake by myeloid cells compared with other fungal species. These points and additional studies have been cited and discussed in the revised manuscript.

Major comment 4: *In the second section of the results, authors state on line 110 "CNS-resident macrophages, including inflammatory macrophages...". This concept is not clear. Upon neuroinflammation*

microglia are activated and trigger a neuroinflammatory process which recruits "for help" peripheral macrophages. These immune cells are not CNS-resident. Authors should clarify this concept.

Response: We have updated the text to make clear that the inflammatory macrophages are not resident and are recruited upon infection.

Major comment 5: *In addition, the experimental design is confusing. In the first section of the results, authors have performed CX3CR1hiCD45int microglia depletion, and such microglia depletion led to a significantly reduced brain fungal burden. In the second section of the results, authors wanted to perform "specific microglia depletion while leaving other CNS-resident macrophages intact". Authors should clarify the class of macrophages (CNS-resident) present in the brain, and what is their role and % of presence in the brain, assuming that microglia are known to be the primary resident brain macrophages.*

Response: Our data presented in Figs 1 and 2 show that microglia are the most numerous myeloid cell type and resident macrophage in the brain. 'Macrophages' in these figures refers to a heterogeneous population that likely contains inflammatory macrophages in addition to rarer populations of border macrophages (see Fig S2 for our gating strategy). While these rare brain-resident macrophages (e.g. perivascular macrophages, choroid plexus macrophages, meningeal macrophages) are independent populations, these can be difficult to discern from recruited macrophages using flow cytometry.

Major comment 6: *In Figure 2 authors assessed that the absence of microglia, but normal numbers of meningeal macrophages and other macrophage populations, led to significantly reduced brain fungal burden. An important control would be to have mice with depleted meningeal macrophages and other CNS-resident macrophages but normal number of microglia and assess if brain fungal burden is not reduced. Alternatively, as suggested above, microglia and other CNS-resident macrophages should be isolated from mouse brains and phagocytosis capacity towards fungal infection should be analyzed in vitro. Comparison of the phagocytic capacity of microglia and CNS-resident macrophages can be provide an explanation of the scenario observed in vivo using Sall1CreERCsf1rflox mice.*

Response: We agree with the reviewer that a model in which CNS macrophages could be depleted while leaving microglia intact would be very useful. However this is currently not possible, since there are no known Cre-lines that can target these populations without affecting microglia (to our knowledge). However, we do now include data comparing microglia and macrophage capacity for restricting fungal growth (see new Fig 6 and response above).

Major comment 7: *The third section of results set a very important new concept in the neuro-infection field, which is that microglia are not the primary immune cells triggering the neuroinflammatory response that leads to the recruitment of peripheral immune cells into the brain. If this concept is true, or partially true, authors should also use mice with depleted CNS-resident macrophages and intact microglia. Alternatively, primary microglia and CNS-resident macrophages should be isolated from mouse brains and release of neuroinflammatory cytokines and chemokines should be analyzed upon infection in vitro with C. neoformans. Such experiment would provide a proof that microglia or other CNS-resident macrophages are involved, or not, in the neuroinflammatory response during fungal infections.*

Response: As outlined in the response to the previous comment, to set up the macrophage-depletion model suggested is not currently possible. We now include new data examining production of CXCL9 from microglia and macrophages in new Fig 7, which suggest a greater role for macrophages in the neuroinflammatory response (see also response to comment 11).

Major comment 8: *The analysis of intracellular fungal residence within microglia is not accurate. The simple observation of microbes inside microglia or macrophages is not an absolute indication of either microbial survival or killing by the immune cell. Visualization of pathogens inside immune cell should be accompanied by a co-staining of an autophagolysosomal marker, such as LC3B, to have an indication if the intracellular microbes are degraded or undergoing a degradation path. If such co-localization is not detected, then it is justifiable to conclude that intracellular microbes are viable inside immune cells. Only the detection*

of microcolonies (more frequent for bacteria) inside immune cells is an indication of microbes growing inside immune cells. Alternatively, to assess in a quantification manner the capacity of microbes to grow inside immune cells, a phagocytosis (also known as gentamicin-protection assay, for bacterial infections, the same can be performed for fungal infections using an anti-fungal agent) should be performed in vitro. After gentamicin/anti-fungal treatment, all extracellular microbes are killed and immune cells with intracellular microbes can be lysed at different time-points and plated. If the count of microbial colonies is increasing over time, it means that microbes are growing inside immune cells.

Response: We now include new data that measured fungal residence in microglia ex vivo, and have compared this to other myeloid cell types in the brain (see new Fig 6). The phenomenon of intracellular survival of this fungus in macrophage phagosomes is well known and there are many published studies demonstrating this. We have included additional citations on this in the revised manuscript (lines 148-149).

Major comment 9: *Authors have focused only on the capacity of intracellular fungi to survive and grow inside microglia, however they have not considered to assess, on the other hand, the phagocytic capacity of microglia. Can microglia and CNS-resident macrophages overexpress phagocytic markers (like TGFbeta) upon fungal infection?*

Response: The receptors and mechanisms of phagocytosis between different myeloid cell types in the brain for *C. neoformans* is an interesting question raised by the reviewer, but one we feel is out of scope for the current study. In the current work, we focus on the consequences occurring post-phagocytosis for host and fungus, but have added additional text to the revised discussion highlighting the questions raised by the reviewer.

Major comment 10: *Authors have shown that C. neoformans is protected from copper starvation within microglia. The results shown are convincing, however, this result section seems to point out that presence of microglia is helping C. neoformans to survive and grow well in the brain thanks to the good levels of copper inside microglia. This can be a valid explanation of the reduced brain fungal burden when microglia are absent. Authors should assess whether C. neoformans is protected from copper starvation also in other CNS-resident macrophages, using a mouse model in which microglia are depleted and CNS-resident macrophage levels are not altered. Is this scenario specific for microglia only? Authors have not excluded that C. neoformans is protected from copper starvation also in other CNS-resident macrophages.*

Response: We now provide new data in Fig 6a that compares the fungal copper starvation response in microglia with other cell types in the brain, which shows that macrophages are able to restrict copper from intracellular fungi (also see responses to reviewer 1).

Major comment 11: *In the last result section, authors state that “IFNgamma also has been shown to modulate copper concentrations within macrophage phagosomes...”. However, in this result section, authors focus only on microglia without considering other CNS-resident macrophages. Also, authors use the term “phagosome”, and the concept of phagocytosis usually implies that phagocytosed particles (microbes too) are eliminated by immune cells. Two are the unclear aspects of this result section: 1. Why authors have not analyzed CNS-resident macrophages too for what concerns the copper restriction induced by IFN gamma 2. Could the copper levels inside microglia/macrophages a way used by the immune cells to attract C. neoformans and phagocytose fungi? It’s important to be not only in the “mind” of the microbes, but also the host cell perspective should be considered.*

Response: As outlined in previous responses, we now include data comparing microglia with macrophages for multiple parameters. We also now include new data comparing downstream IFN γ responses within microglia and macrophages, using intracellular staining for CXCL9, a canonical IFN γ -regulated chemokine (this strategy was used to maintain our flow cytometry panel to distinguish between microglia and macrophages while remaining compatible with fixation/permeabilisation steps), which is presented in new Fig 7. That experiment showed that macrophages have increased responsiveness to IFN γ during infection, which may explain the difference in fungal copper starvation responses between microglia and macrophages. This new data and its implications are discussed in the revised manuscript. We think the reviewer raises an

interesting point regarding attraction of fungi to host cells via metals. However, if this was the case, we think we would have observed greater levels of uptake of the fungus by host cells which was rarely above 10% of the total population. Our ongoing studies into the molecular mechanisms regulating intracellular copper access in different cell types during infection will hopefully yield new insights into how manipulation of this axis affects host-fungal interactions in the brain.

Minor comments:

1) How have authors assessed the levels of microglia after cell depletion post tamoxifen treatment without any infection prior to performing the experiments shown in the manuscript? Has depletion of microglia been always consistent?

Response: All of our depletion models showed consistent depletion post-infection. We have now included baseline data in Fig S2 to demonstrate the level of depletion achieved before infection.

2) Why have authors taken in consideration meningeal macrophages? Infection of the mice was done intravenously, so fungi have entered the brain via the blood through the BBB. Meningeal macrophages can be important in case of an intracisternal route of infection, but not primarily in this case. Have authors performed staining to assess by microscopy if meningeal macrophages are activated? Furthermore, meninges are extremely difficult to collect from mice. Why haven't authors considered pericytes, the perivascular macrophages of the BBB, which should definitely have a more important role in this study since the infection is blood-borne.

Response: We examined all brain myeloid populations to gain a sense of the potential wide-reaching effects of the depletion strategy used in each model. The reviewer raises an interesting point about the role of the meninges. We have actually performed an in depth analysis of this tissue and are developing a manuscript on this, in which we will provide microscopy and immunological/transcriptomic analysis of meningeal macrophages. This has revealed new roles for macrophages at this site in the pathogenesis of this infection (to be included in our future manuscript). Only our Sall1-CreER-Csf1rflox model is able to leave the perivascular macrophages intact, but these do not appear to play a significant role based on the fungal burden data obtained with these mice (at least within the time frame studied).

REVIEWER COMMENTS

Reviewer #1 (Remarks to the Author):

The manuscript focused on the role of microglia in creating an improved survival niche in the CNS for the fungal CNS pathogen, *Cryptococcus neoformans* has been submitted after revisions. The authors provided additional data and made multiple changes to the manuscript, however, the manuscript, in its current form is still not ready for publication due to the speculative nature of the proposed mechanism. More studies to ascertain whether copper (Cu) is the key facilitator or if other contributing factors are at play. Without such investigation, attributing the increased fungal growth solely to higher Cu concentration, suggested by the reporter strain responses, remains speculative. One alternative way microglia might help in this model could be simply "intracellular" hiding of the fungus from more fungicidal neutrophils, which would be Cu-independent.

A potential approach to testing the role of Cu could involve utilizing a Cu-non-sensitive strain of *C. neoformans* (perhaps one that overexpresses copper transporters) or testing this mechanism *in vitro* by manipulating the copper concentration, which may offer a clearer understanding. This approach might help to determine whether copper is the primary factor promoting fungal growth, or if other factors are contributing as well. Without such validation, the manuscript's conclusions are premature.

Specific comments:

1) The manuscript is based on the assumption that microglia are a copper reservoir in the CNS with 2-3 orders of magnitude greater copper concentration than in the extracellular milieu. The cited reference supporting this statement does not measure Cu in the microglia and the authors state that they could not measure Cu in the microglia because it was below the limits of detection. However, they can measure it in the whole brain. These explanations seem not to add up, and since the only data pertaining Cu concentration are based on the expression of the reporter gene by the strain, it is unclear if other factors cannot modify the expression of this gene apart from sensing Cu concentration.

2) The studies requested by reviewers adding back Cu were not conducted due to possible cytotoxicity issues, however, the range of Cu concentrations the reporter responds to (50nM) is way below their overall Cu concentration in the CNS (3,000nM) gives a lot of room to manipulate Cu. An alternative, approach would be to decrease microglia Cu concentration via means other than IFN- γ . IFN- γ is known to increase a variety of fungicidal mechanisms in phagocytes and the observed change in the reporters of MFI, while significant is subtle (about 40%) compared to the 3-fold change in the extracellular environment. So other methods manipulating Cu concentration are needed to provide more solid evidence.

3) The data pertaining to the role of microglia is "spotty" in terms of the response of the individual strains of *C. neoformans* and timing while the authors state that the depletion of microglia has a "protective effect" in the abstract. The highly virulent strain H99 benefits from the presence of microglia during the early stage of the infection but two other strains tested do not. The authors offer some explanations that it is due to the variable level of virulence, however, the Dctr1/Dctr4 strain has highly reduced virulence and it still benefits from the presence of microglia. I think the study shows that microglia, can be conditionally beneficial for the pathogen but making a generalized conclusion that it is protective for fungi in the CNS is a stretch and is misleading in the light of the data shown.

Other comments:

Fig 6 No Stats on the killing experiment. Data shows that the Cu-restriction is limited to extracellular milieu and macrophages but not neutrophils or monocytes. Neutrophils, however, are best at killing cryptococcus. The correlation between the level of Cu restriction and fungicidal properties of cells does not hold here. Again what is the Cu growth limiting for cryptococcus? Also, the new data indicates microglia have the lowest levels of other microelementary metals. Reporter strain response shows similar levels to neutrophils and monocytes. Together, these findings seem to be counterintuitive to the proposed role for Cu in the microglia.

Reviewer #2 (Remarks to the Author):

This is a resubmission of the study entitled "Microglia protect fungi against copper starvation and promote early brain infection." The authors responded well to comments by previous reviewers and added new data. Their microglia depletion was convincing and nicely performed with multiple approaches. However, the major concern of this manuscript is that the title and abstract only focus on limited positive results (See the following paragraph for this). Thus, this presentation overstates positive results alone and is quite misleading.

Here is a little more explanation about the overstatement issue -- The data showed that only the H99 strain of *Cryptococcus neoformans* was benefited from microglia in fungal burdens, and other strains do not show any difference with or without microglia. Here, the impact of microglia depletion on H99 is also relatively small, although the data showed statistical significance. The effect of microglia in H99 infection also seems to be transient because two out of three readouts do not show a difference without microglia (Fig. 4e). Once it gets to the brain inflammation, including peripheral immune cell recruitment, even H99 brain infection does not make a difference without microglia. All the data suggest a limited impact of microglia on *C. neoformans* infection in general. Thus, the title "Microglia protect fungi against copper starvation and promote early brain infection" is quite misleading. The abstract is also misleading and does not reflect negative data at all. This is a serious concern.

- Mouse survival is not examined. This is an ultimate readout to determine whether microglia are detrimental or beneficial to the hosts.
- The authors described that microglia provide a safe haven for *C. neoformans*. Although this is not incorrect, it is an overstatement. This particular evaluation on copper is limited to the strain H99-derived strains. (The Ctr1Δ/ctr4Δ appears to be made on the H99 background based on the referred article by Sun et al. Nat Comm, 2014). Here, H99 is the only strain, among other tested strains, which are impacted by microglia in brain CFU. Thus, the microglia-specific impact favorable for *C. neoformans* cannot be generalized.
- Relating to the interpretation of microglia providing an intracellular safe space for *C. neoformans*, Fig. 6d shows that 50 % of microglia kill *C. neoformans*. Also, the sentence starting in line 300 states that "microglia do not appear to have a greater abundance of trace metals than other cell," based on the data in Fig. 6b.
- The idea of copper involvement has already been reported. "Reciprocal functions of *Cryptococcus neoformans* copper homeostasis machinery during pulmonary infection and meningoencephalitis (Sun et al. 2014. Nat Comm)." So this aspect is not considered to be novel.
- Line 239 states, "*C. neoformans* strain lacking CTR4 and the related importer CTR1 have significantly reduced capacity to infect the mouse brain." For this statement, Fig. 5b needs the control group, where the WT-H99 strain infects WT mice.
- Line 244 "We next compared expression of CTR4 by yeast cells inside microglia with extracellular yeast in the brain in vivo,..." This sentence needs to be rephrased because it is not clear what "yeast cells inside microglia with extracellular yeast" means.
- Approach shown in Fig. 4a, b is great, but the data there are not convincing. Particularly, it is very difficult to identify yeasts outside the lesion "holes" to be intracellular or extracellular without marking the cell membrane.
- CXCL9 expression cannot be used to indicate IFN γ signaling because macrophages and microglia may not share the same IFN γ receptor signaling.
- Fig. 7a data needs isotype control. It is still possible that macrophages may cause more background staining than microglia, particularly because the positive population came from the shift of one population seen in the flow cytometry image.

Reviewer #3 (Remarks to the Author):

Excellent work of revision by the authors. All my questions were properly addressed. Only one last remark. Concerning the new data in Fig 6, can authors describe how exactly microglia, macrophages, monocytes, and neutrophils were sorted by FACS? I did not find this information in the Methods section.

Reviewer #1 (Remarks to the Author):

Major comment 1: *The manuscript focused on the role of microglia in creating an improved survival niche in the CNS for the fungal CNS pathogen, Cryptococcus neoformans has been submitted after revisions. The authors provided additional data and made multiple changes to the manuscript, however, the manuscript, in its current form is still not ready for publication due to the speculative nature of the proposed mechanism. More studies to ascertain whether copper (Cu) is the key facilitator or if other contributing factors are at play. Without such investigation, attributing the increased fungal growth solely to higher Cu concentration, suggested by the reporter strain responses, remains speculative. One alternative way microglia might help in this model could be simply “intracellular” hiding of the fungus from more fungicidal neutrophils, which would be Cu-independent.*

Response: In the revised version of the manuscript, we have updated the title, rewritten the abstract and slightly re-organised the data to emphasise the key finding of the paper, which is that microglia do not provide protection against this infection in contrast to other types of infections. Our work demonstrates that microglia, while not protective, may contribute towards the development of heterogeneity within the fungal population in the brain, which is at least partly driven by differing concentrations of copper. Importantly, the CTR family comprises a collection of extensively studied proteins responsible for the transport of copper ions, exhibiting a high degree of conservation across many organisms ranging from fungi to mammals. Numerous prior investigations on *C. neoformans* CTR genes have provided comprehensive evidence indicating that the promoter of these genes exclusively exhibits responsiveness to copper levels (PMID: 23498952; PMID: 31932719; PMID: 20444417). Moreover, strains with disrupted CTR genes have been observed to exhibit growth solely in the presence of copper supplementation within the growth medium (PMID: 21819456). This means that our studies with the CTR4-reporter strain can help accurately assess regulation of copper responses by the fungus within the host during meningitis. We appreciate the hypothesis put forward by the reviewer that microglia may have served as a protective environment for *C. neoformans* cells against neutrophil activity. However, this observation does not offer an explanation for the ability of extracellular *C. neoformans* to detect the lack of copper. Moreover, it should be noted that the activation of and the respiratory burst within neutrophils are activities that rely on the presence of copper. Taken together, we feel our work demonstrating heterogeneity of copper responses by *C. neoformans* in the brain, and the role microglia play in driving this, is an important finding and highlights the significance of considering micro-niches in the analysis of fungal responses within the host. We ultimately agree with the reviewer that there is more work to be done on the role of copper, which will require several more experiments and additional manuscripts.

Major comment 2: *A potential approach to testing the role of Cu could involve utilizing a Cu-non-sensitive strain of C. neoformans (perhaps one that overexpresses copper transporters) or testing this mechanism in vitro by manipulating the copper concentration, which may offer a clearer understanding. This approach might help to determine whether copper is the primary factor promoting fungal growth, or if other factors are contributing as well. Without such validation, the manuscript's conclusions are premature.*

Response: We have now performed the suggested experiment with a copper insensitive strain and include this data in Fig 7. We infected wild-type and microglia-

deficient mice with a strain that over-expresses CTR4 and found no difference in brain fungal burdens, in contrast to the experiments that used a *C. neoformans* strain that is unable to adapt to copper deficient environments (CTR1/4-deficient). Those experiments show that the role of microglia is influenced by the ability of the fungus to adapt to copper deficiency.

Major comment 3: *The manuscript is based on the assumption that microglia are a copper reservoir in the CNS with 2-3 orders of magnitude greater copper concentration than in the extracellular milieu. The cited reference supporting this statement does not measure Cu in the microglia and the authors state that they could not measure Cu in the microglia because it was below the limits of detection. However, they can measure it in the whole brain. These explanations seem not to add up, and since the only data pertaining Cu concentration are based on the expression of the reporter gene by the strain, it is unclear if other factors cannot modify the expression of this gene apart from sensing Cu concentration.*

Response: The majority of copper is protein-bound in the brain and CSF, hence why intracellular concentrations are expected to be higher than extracellular concentrations. Measurements of copper in human CSF are much lower than in serum, but copper may increase during infection or disease in line with increasing protein in the CSF (PMC:3926505). Copper is not only in microglia, but also in other cells in the brain such as the more numerous astrocytes and oligodendrocytes. This is probably why we can measure copper in whole tissue (as we can detect it across multiple cell types) but are limited when we isolate specific cells which are more difficult to extract from tissue. Detecting copper in other ways (for example, with a probe) is possible but these tools are not commercially available and we do not have the chemistry skillset to generate them in-house. We and others have shown that CTR4 expression is determined by external copper concentrations, which occurs via the copper-sensitive transcription factor CUF1 (also see response above). We have removed statements from the manuscript that indicated differences in CTR4 expression are only due to copper concentration since we did not specifically test this. We have also updated the discussion to make our main conclusion clearer; that fungal populations in the brain are heterogeneous and intracellular environments have a significant influence over this.

Major comment 4: *The studies requested by reviewers adding back Cu were not conducted due to possible cytotoxicity issues, however, the range of Cu concentrations the reporter responds to (50nM) is way below their overall Cu concentration in the CNS (3,000nM) gives a lot of room to manipulate Cu. An alternative, approach would be to decrease microglia Cu concentration via means other than IFN-g. IFN-g is known to increase a variety of fungicidal mechanisms in phagocytes and the observed change in the reporters of MFI, while significant is subtle (about 40%) compared to the 3-fold change in the extracellular environment. So other methods manipulating Cu concentration are needed to provide more solid evidence.*

Response: The comparison between in vitro cultured cell lines and primary cells within their in vivo environment is not completely fair. Cells within tissues are supported and interact with other cells, the extracellular matrix and soluble factors that can't be replicated in vitro, which means that the effects of adding exogenous metals to cultured cell lines can be difficult to interpret and will not always align with observations made in vivo. We don't yet know the mechanisms that regulate copper levels within cells during intracellular fungal infection to be able to manipulate copper concentrations with multiple independent methods. Our lab is currently attempting to answer these questions however these studies are out of

scope of the current work where we focus on the impact of microglia depletion on control of *C. neoformans* infection. The revisions to the structure of the paper and abstract have been made to make this emphasis clearer.

Major comment 5: *The data pertaining to the role of microglia is "spotty" in terms of the response of the individual strains of C. neoformans and timing while the authors state that the depletion of microglia has a "protective effect" in the abstract. The highly virulent strain H99 benefits from the presence of microglia during the early stage of the infection but two other strains tested do not. The authors offer some explanations that it is due to the variable level of virulence, however, the Dctr1/Dctr4 strain has highly reduced virulence and it still benefits from the presence of microglia. I think the study shows that microglia, can be conditionally beneficial for the pathogen but making a generalized conclusion that it is protective for fungi in the CNS is a stretch and is misleading in the light of the data shown.*

Response: We apologise to the reviewers for not revising the abstract in the previous submission – this was a genuine oversight and has now been corrected and revised to reflect all of the new data. As outlined in previous responses, we have revised the paper to make the main conclusions clearer, and have moved some of the data on strains and later time-points into the earlier figures as we think this will strengthen the message that microglia are not protective for *C. neoformans* infection, unlike other infections.

Major comment 6: *Fig 6 No Stats on the killing experiment. Data shows that the Cu-restriction is limited to extracellular milieu and macrophages but not neutrophils or monocytes. Neutrophils, however, are best at killing cryptococcus. The correlation between the level of Cu restriction and fungicidal properties of cells does not hold here. Again what is the Cu growth limiting for cryptococcus? Also, the new data indicates microglia have the lowest levels of other microelementary metals. Reporter strain response shows similar levels to neutrophils and monocytes. Together, these findings seem to be counterintuitive to the proposed role for Cu in the microglia.*

Response: Statistics have been added to the killing experiment. Our data show that while copper starvation occurs in the CNS, this does not equate to more killing and we think this is because *C. neoformans* can adapt to copper restriction (i.e. by upregulating genes such as CTR4). Knocking out these genes in *C. neoformans* reduces virulence and growth but this may be partially overcome by the presence of microglia, while absence of microglia does not affect a strain that is insensitive to copper-deficiency (see our experiments in Fig 7 with the CTR-deficient and over-expression strains). These data mean that adaptation to copper starvation is an important defence of the fungus within the brain and may help subvert immune cell killing, and there is not a simple correlation between copper access within immune cells and their fungal killing capacity. All of these points have been made in the revised manuscript.

Reviewer #2 (Remarks to the Author):

Major comment 1: *This is a resubmission of the study entitled "Microglia protect fungi against copper starvation and promote early brain infection." The authors responded well to comments by previous reviewers and added new data. Their microglia depletion was convincing and nicely performed with multiple approaches. However, the major concern of*

this manuscript is that the title and abstract only focus on limited positive results (See the following paragraph for this). Thus, this presentation overstates positive results alone and is quite misleading.

Response: As outlined in responses to reviewer 1, we have now rewritten the abstract, given the paper a new title and slightly re-organised data to address these concerns. The revised paper emphasises the lack of a role for microglia in infection control, which we believe is still striking in contrast to the protective role for these cells in other types of infections and therefore an important finding for the field.

Major comment 2: *Here is a little more explanation about the overstatement issue -- The data showed that only the H99 strain of Cryptococcus neoformans was benefited from microglia in fungal burdens, and other strains do not show any difference with or without microglia. Here, the impact of microglia depletion on H99 is also relatively small, although the data showed statistical significance. The effect of microglia in H99 infection also seems to be transient because two out of three readouts do not show a difference without microglia (Fig. 4e). Once it gets to the brain inflammation, including peripheral immune cell recruitment, even H99 brain infection does not make a difference without microglia. All the data suggest a limited impact of microglia on C. neoformans infection in general. Thus, the title "Microglia protect fungi against copper starvation and promote early brain infection" is quite misleading. The abstract is also misleading and does not reflect negative data at all. This is a serious concern.*

Response: Please see responses to reviewer 1.

Major comment 3: *Mouse survival is not examined. This is an ultimate readout to determine whether microglia are detrimental or beneficial to the hosts.*

Response: Mouse survival experiments are strongly discouraged in the UK due to 3Rs implications. The model we have used in the study is 100% lethal which also makes it difficult to discern differences in survival between groups. We would not expect to see a difference between wild-type and microglia-depleted mice since the fungal brain burden equalises between these two groups as infection progresses. We have previously used body weight as a measure of general health in this model, although we typically find that mice only lose weight (>15%) upon development of meningitis or other clinical symptoms (which we aim to avoid in our model to keep animals within a moderate severity limit). For the reviewer, we include weight data below from an example experiment comparing wild-type and microglia-depleted mice (following intravenous infection, as outlined in methods of our paper) which shows no significant difference across multiple mice.

Major comment 4: *The authors described that microglia provide a safe haven for C. neoformans. Although this is not incorrect, it is an overstatement. This particular evaluation on copper is limited to the strain H99-derived strains. (The Ctr1Δ/ctr4Δ appears to be made on the H99 background based on the referred article by Sun et al. Nat Comm, 2014). Here, H99 is the only strain, among other tested strains, which are impacted by microglia in brain CFU. Thus, the microglia-specific impact favorable for C. neoformans cannot be generalized.*

Response: We have softened the text throughout to make clear that the data on CTR4 expression reflects how microglia (and other host cells) may influence the development of heterogeneous fungal populations in the brain, of which copper access may be contributing. See also responses to reviewer 1.

Major comment 5: *Relating to the interpretation of microglia providing an intracellular safe space for C. neoformans, Fig. 6d shows that 50 % of microglia kill C. neoformans. Also, the sentence starting in line 300 states that “microglia do not appear to have a greater abundance of trace metals than other cell,” based on the data in Fig. 6b.*

Response: See earlier responses; the text has been revised to soften these statements and emphasise the lack of role for microglia.

Major comment 6: *The idea of copper involvement has already been reported. “Reciprocal functions of Cryptococcus neoformans copper homeostasis machinery during pulmonary infection and meningoencephalitis (Sun et al. 2014. Nat Comm).” So this aspect is not considered to be novel.*

Response: As noted by the reviewers, there is limited research investigating how *C. neoformans* detects copper deficit during brain infection, particularly from the microbial standpoint, with a lack of comprehensive understanding regarding the corresponding reactions and mechanisms in mammals. There are several unresolved inquiries pertaining to the mechanisms underlying copper deficiency in the brain. Additionally, it remains unclear how the immune system of the brain employs copper as an anti-fungal defence mechanism during cryptococcal meningitis. While the reviewer is correct that the idea of copper starvation by *C. neoformans* in the brain has been previously outlined, there are several crucial connections between brain immunity and copper balance that remain unexplored. Our study is the inaugural investigation to demonstrate that these reactions are not uniformly manifested throughout the fungal population, but rather exhibit variations depending on the

specific environmental niche inhabited by the fungus. This study represents the initial investigation into the intricate mechanisms behind copper deprivation in the context of brain infection, while also highlighting the significant contribution of microglia in regulating copper balance during cryptococcal meningitis. This is an important finding with significance for the field's future trajectory. The analysis of yeast inside whole tissues as a uniform population overlooks crucial distinctions that may have consequential implications for virulence and infection control. Therefore, it is essential to delve further into these variations and conduct more investigations.

Major comment 7: Line 239 states, "*C. neoformans* strain lacking *CTR4* and the related importer *CTR1* have significantly reduced capacity to infect the mouse brain." For this statement, Fig. 5b needs the control group, where the WT-H99 stain infects WT mice.

Response: The WT data is already included in the manuscript in Fig 2g; we added a statement to the revised manuscript to make this easier to find (line 349). We have also included brain burden data below from wild-type mice infected with the same dose of H99 (the parental control for the mutant) and the *CTR1/4* dKO strain. Based on that data, we used a higher dose of the *CTR1/4* dKO strain in subsequent experiments since brain burdens were very low. A similar high dose of H99 would have been quickly lethal hence why it is not shown alongside the mutant data.

Major comment 8: Line 244 "We next compared expression of *CTR4* by yeast cells inside microglia with extracellular yeast in the brain in vivo,..." This sentence needs to be rephrased because it is not clear what "yeast cells inside microglia with extracellular yeast" means.

Response: This sentence has been rephrased.

Major comment 9: Approach shown in Fig. 4a, b is great, but the data there are not convincing. Particularly, it is very difficult to identify yeasts outside the lesion "holes" to be intracellular or extracellular without marking the cell membrane.

Response: We agree with the reviewer that identifying intracellular yeast is difficult and this is why have broadly scored between yeast in the lesions (extracellular) and outside of the lesions (non-lesion growth). We have updated the label in our graphs (Fig 1 and Fig 4) to make this approach in our analysis clearer.

Major comment 10: *CXCL9 expression cannot be used to indicate IFN γ signaling because macrophages and microglia may not share the same IFN γ receptor signaling.*

Response: The reviewer is correct that microglia and macrophages may differ in their IFN γ signalling, and indeed our data alludes to this. However, we think it is the downstream biological effects that are important for the conclusions in the current work. This will be an important future direction to discern why IFN γ has differential effects on different subsets of myeloid cells and we have added some extra discussion on this point to the revised manuscript.

Major comment 11: *Fig. 7a data needs isotype control. It is still possible that macrophages may cause more background staining than microglia, particularly because the positive population came from the shift of one population seen in the flow cytometry image.*

Response: Staining of the isotype control on microglia and macrophages is now shown in Fig S4. There was no change in the shift between these two populations.

Reviewer #3 (Remarks to the Author):

Major comment 1: *Excellent work of revision by the authors. All my questions were properly addressed. Only one last remark. Concerning the new data in Fig 6, can authors describe how exactly microglia, macrophages, monocytes, and neutrophils were sorted by FACS? I did not find this information in the Methods section.*

Response: We have now added the methods section for these experiments to the manuscript.

REVIEWERS' COMMENTS

Reviewer #1 (Remarks to the Author):

The authors have addressed my comments regarding the discrepancy between findings and conclusions in the revised manuscript.

Reviewer #2 (Remarks to the Author):

This is the second revision. The authors added major text edits in the draft, which reflects reviewers' previous critiques. The current version appears to describe the experimental results more accurately than the previous versions.